# Giant electrocaloric materials energy efficiency in highly ordered lead scandium tantalate

Youri Nouchokgwe [1,2✉], Pierre Lheritier[1], Chang-Hyo Hong[3], Alvar Torelló [1,2], Romain Faye[1], Wook Jo [3], Christian R. H. Bahl [4] & Emmanuel Defay [1✉]

Electrocaloric materials are promising working bodies for caloric-based technologies, suggested as an efficient alternative to the vapor compression systems. However, their materials efficiency defined as the ratio of the exchangeable electrocaloric heat to the work needed to trigger this heat remains unknown. Here, we show by direct measurements of heat and electrical work that a highly ordered bulk lead scandium tantalate can exchange more than a hundred times more electrocaloric heat than the work needed to trigger it. Besides, our material exhibits a maximum adiabatic temperature change of 3.7 K at an electric field of 40 kV cm$^{-1}$. These features are strong assets in favor of electrocaloric materials for future cooling devices.

[1] Materials Research and Technology Department, Luxembourg Institute of Science and Technology, Belvaux, Luxembourg. [2] University of Luxembourg, Esch-sur-Alzette, Luxembourg. [3] School of Materials Science and Engineering, Ulsan National Institute of Science and Technology, Ulsan, South Korea. [4] Department of Energy Conversion and Storage, Technical University of Denmark, Lyngby, Denmark. ✉email: youri.nouchokgwe@list.lu; emmanuel.defay@list.lu

F ridges and air conditioning consume fifteen percent of the world's energy[1]. Looking for an alternative to the almost exclusively used vapor compression system is a crucial technological challenge as one needs to suppress the use of greenhouse gases and improve energy efficiency as much as possible[2,3]. Recent findings have demonstrated that electrocaloric (EC) devices represent a more and more credible alternative[4–6].

Imposing caloric-based cooling requires high performance materials, which are also energy efficient[7–9]. This efficiency can be intrinsic but also extrinsic. Indeed, in the case of EC materials, it has been shown that the overall efficiency of a heat pump could be improved threefold by recycling the electrical charges used to trigger cooling in its EC modules[10]. On the contrary, the intrinsic efficiency of EC materials has been largely overlooked even though it plays a part in the efficiency of the prototype[8].

Here we use the parameter materials efficiency ($\eta_{\mathrm{mat}}$) as the figure of merit to rank caloric materials in terms of their energy efficiency[8,9]. Intrinsic to any material, $\eta_{\mathrm{mat}}$ quantifies how much heat $Q$ - induced by the entropy change, which is triggered by the caloric effect—a material can exchange by heat transfer with its surroundings, with respect to the work $W$ needed to drive this caloric effect. Hence, $\eta_{\mathrm{mat}}$ is simply $|Q/W|$[7–9]. This dimensionless quantity only depends on material properties and is independent of the parameters of any cooling device (sink and load temperature, cooling power). It is worth noting that this definition enables $\eta_{\mathrm{mat}}$ being larger than 1. Indeed, it is often much larger. This expresses the fact that one can trigger large heat exchange (in Joules) with a much smaller driving work (in Joules as well) especially close to phase transitions. $\eta_{\mathrm{mat}}$ should not be confused with scaled efficiency, which is the ratio of a device's Coefficient of Performance (COP) to the Carnot COP and is always less than 1. The study of $\eta_{\mathrm{mat}}$ published in 2013 aimed to compare EC ceramics films ($\eta_{\mathrm{mat}} = 3$) to polymers films ($\eta_{\mathrm{mat}} = 7.5$)[7]. In 2015, Moya et al. compared magnetocaloric (MC), EC, and mechanocaloric (mC) materials using the indirect method based on Maxwell's thermodynamic relations[8]. They showed that MC materials are an order of magnitude more efficient than EC thin films.

Lead Scandium Tantalate Pb(Sc$_{1/2}$Ta$_{1/2}$)O$_3$, short form PST[11–16] is one of the most promising EC materials, already utilized to build EC heat pumps[4,5,17–19]. It exhibits a first-order ferroelectric to paraelectric (PE) phase transition near room temperature which makes it attractive for air conditioning. This transition can also be driven with an electric field leading to the EC effect, which is predominant at the material's transition temperature[13]. The largest EC adiabatic temperature change $\Delta T_{\mathrm{adiab}}$[13] reported so far in bulk PST is 2.3 K at 50 kV cm$^{-1}$. Lately, Nair et al.[20] measured a larger $\Delta T_{\mathrm{adiab}}$ of 5.5 K in PST Multi-Layer Capacitors (MLCs) when driven supercritically at

290 kV cm$^{-1}$. In addition, we showed that an EC prototype built using 128 PST MLCs can reach a giant temperature span of 13 K[4]. These previous works demonstrate that PST remains one of the best candidates for EC cooling. However, a crucial parameter largely unknown is its materials efficiency.

Consequently, here we report on the materials efficiency of highly ordered bulk PST, by direct measurements of heat and electrical work. Heat is measured using Differential Scanning Calorimetry (DSC) and thermometry with an infrared (IR) camera. Voltage and current needed to charge PST ceramics are recorded simultaneously to extract electrical work, needed to obtain PST's $\eta_{\mathrm{mat}}$. We reveal that highly ordered bulk PST exhibits a maximum $\eta_{\mathrm{mat}}$ of 128, as high as the best caloric materials[8,9]. Besides, our bulk PST reaches a $\Delta T_{\mathrm{adiab}}$ as large as 3.7 K at 40 kV cm$^{-1}$ due to its higher degree of ordering. We also show that bulk PST can be four times more efficient than Gadolinium (Gd) in some specific conditions. Therefore, bulk PST is potentially an excellent caloric material due to its large temperature change and materials efficiency.

## Results

**Electrocaloric effect in PST**. The fabrication of PST is described in the "Methods" section.

X-Ray diffraction (XRD) measurements were carried out to determine the degree of B-site cation order $\Omega$, found to be 0.98 in PST sample 1 (explained in the "Methods" section and XRD scan in Supplementary Fig. 1). Dielectric measurements (Fig. 1a) show a sharp peak at $T_0 = 300$ K on heating with a small hysteresis of 3 K. We record dielectric losses of 0.025 in the ferroelectric (FE) phase, which strongly decreases down to <0.01 in the PE phase. This discrepancy in losses is explained by the presence of domain walls in FE, which are absent in PE. Sharp peaks (Fig. 1b) combined with thermal hysteresis, in zero-field specific heat $C_{\mathrm{p}}$ measurements made by DSC (see Methods section) indicate the presence of a first-order phase transition in PST at the transition temperature $T_0 = 300$ K. A constant $C_{\mathrm{p}}$ of 300 J kg$^{-1}$ K$^{-1}$ which corresponds to the background of the zero-field $C_{\mathrm{p}}$ measurements in Fig. 1b, is measured and in good agreement with the literature[14,15]. The peak at $T_0$ is the latent heat $Q_0$ due to the first-order transition from FE–PE (on heating) or PE–FE (on cooling). The integration of zero-field calorimetric peaks d$Q$/d$T$ (Methods section and Supplementary Fig. 2) yields $Q_0$ of 1031 J kg$^{-1}$ on heating and 890 J kg$^{-1}$ on cooling. From zero-field $C_{\mathrm{p}}$ measurements, we calculate $S'(T)$ the entropy referenced to the entropy at 280 K ($S'(280$ K$) = 0$) far away from the transition (described in the Methods section). We access a large transition entropy change $\Delta S_0 \approx 3.4$ J kg$^{-1}$ K$^{-1}$ on heating and $\Delta S_0 \approx 3.0$ J kg$^{-1}$ K$^{-1}$ on cooling (Fig. 1c).

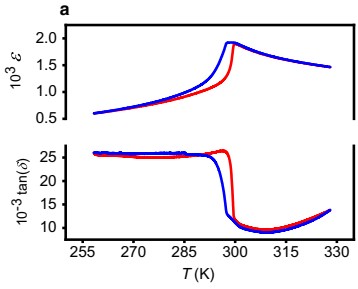
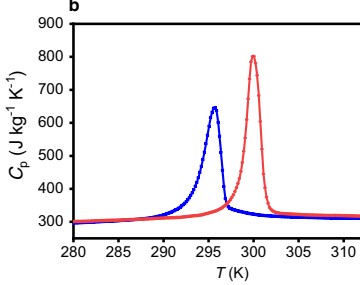
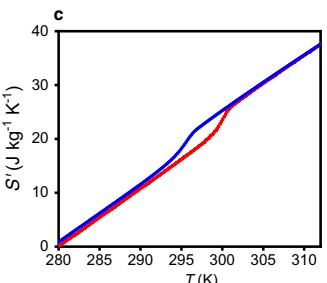

**Fig. 1 Zero-field measurements of PST.** Blue and red curves denote respectively cooling and heating in the three plots. **a** The top plot shows the relative permittivity $\varepsilon$ and the lower plot the loss tangent tan($\delta$), both as a function of temperature $T$. **b** Zero-field specific heat $C_{\mathrm{p}}$ (background value) measurements associated with latent heat $Q_0$ (integral under the first-order phase transition peak at $T_0 = 300$ K on heating and 296 K on cooling). **c** $S'(T) = S(T) - S(280$ K$)$ represents the entropy $S(T)$ referenced to the entropy at 280 K far from $T_0$ with $S'(280$ K$) = 0$.

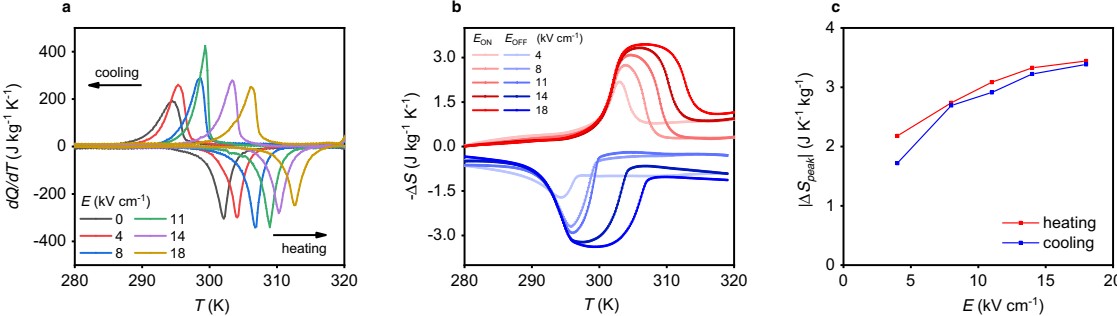

**Fig. 2 Isofield measurements in PST. a** Isofield heat flow measurements d$Q$/d$T$ as a function of temperature $T$ at different electric fields $E$ on heating and cooling. **b** Isothermal entropy change $\Delta S$ obtained from isofield measurements (Methods section) at different electric fields, on applying ($E_{ON}$) and removing ($E_{OFF}$) field. **c** Isothermal entropy change peak $|\Delta S_{peak}|$ as function of electric field $E$, on cooling (blue curves) and heating (red curves).

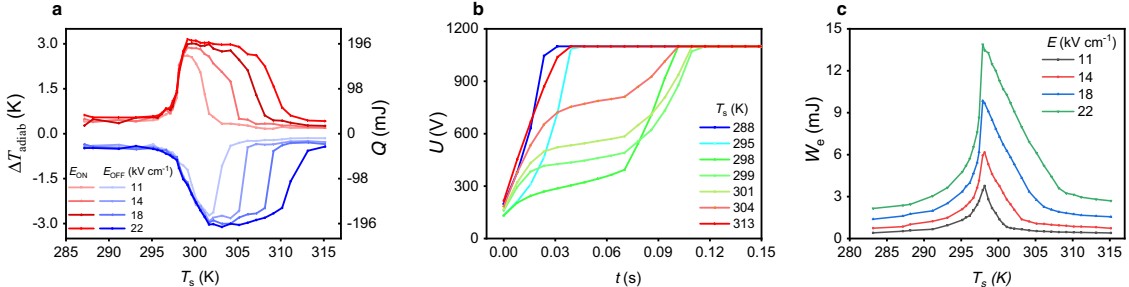

**Fig. 3 PST-heat exchanged $Q$ and electrical work $W_e$. a** Direct $\Delta T_{adiab}$ measurements and electrocaloric heat $Q$ of PST as a function of starting temperatures $T_s$ under application (red curves) and removal (blue curves) of 11, 14, 18, and 22 kV cm$^{-1}$. The protocol carried out to measure $\Delta T_{adiab}$ is detailed in Supplementary Note 9. $Q$ is calculated from directly measured $\Delta T_{adiab}$ and constant background value of $C_p$. **b** Example of charge in PST capacitor at a maximum voltage of 1100 V (22 kV cm$^{-1}$) as a function of time $t$ at selected $T_s$. **c** Electrical work $W_e$ needed to charge a PST capacitor as a function of $T_s$ at different electric fields $E$. $W_e$ is obtained from the data of voltage (Fig. 3b) across PST at constant current.

Isofield measurements by DSC (see Methods) of our PST presented in Fig. 2a. and Supplementary Fig. 3a show a linear shift of the transition temperature $T_0$ toward higher temperatures, and thus a stabilization of the low temperature phase (FE phase) due to the field. A difference of 10 K between the transition temperature at 0 kV cm$^{-1}$ and maximum field applied of 18 kV cm$^{-1}$ is measured and gives an estimated value of the temperature range of the EC effect. The integration of the (d$Q$/d$T$)/$T$ peaks in Fig. 2a. gives a constant $\Delta S_0$ of 3.4 J kg$^{-1}$ K$^{-1}$ with electric field (Supplementary Fig. 3c), which corresponds to the entropy change resulting from fully driving the transition. From isofield measurements and specific heat measurements (Methods section) we built entropy curves $S'$ referenced to 280 K to determine the isothermal field-driven entropy change $\Delta S$ versus temperature (Fig. 2b) and field (Fig. 2c). A maximum isothermal entropy change of 3.44 J kg$^{-1}$ K$^{-1}$ on heating and 3.38 J kg$^{-1}$ K$^{-1}$ on cooling is obtained under the electric field of 18 kV cm$^{-1}$. This value is very similar to $\Delta S_0$ extracted at zero field and proves that the transition can be fully driven at 18 kV cm$^{-1}$. These results are in agreement with previous isofield measurements done on a less ordered bulk ceramic PST[15].

**Electrocaloric exchangeable heat in PST**. IR camera measurements of reversible and reproducible $\Delta T_{adiab}$ performed on a 0.480 cm$^2$ area bulk ceramic 0.5 mm-thick PST sample 1 at different starting temperatures $T_s$ and four electric fields (11, 14, 18, and 22 kV cm$^{-1}$) are presented in Fig. 3a. Under the fast (<0.1 s) application and removal of an electric field at the same starting temperature, positive and negative $\Delta T_{adiab}$ respectively, were measured on three successive electric field cycles using the protocol described in the Methods Section (see also Supplementary

Fig. 12 about adiabatic conditions). Reproducible $\Delta T_{adiab}$ measurements were systematically obtained on these three successive cycles at all investigated fields (Supplementary Fig. 11). Consequently, an average over the three runs is taken to extract $\Delta T_{adiab}$ of PST at a given starting temperature $T_s$ and electric field. At the lowest applied field of 11 kV cm$^{-1}$, we measured a reversible maximum $\Delta T_{adiab}$ of 2.6 K at $T_0 = 299$ K in a temperature range of 2.5 K (Fig. 3a). This temperature range is calculated as the full width at 80% of the maximum $\Delta T_{adiab}$. By increasing the field, we enlarge the peak toward higher temperatures thereby stabilizing the FE phase. This is in agreement with the isofield measurements in Fig. 2a. $\Delta T_{adiab}$ increases up to 3.1 K and the temperature range reaches 8.4 K by doubling the field (22 kV cm$^{-1}$). An even higher $\Delta T_{adiab}$ of 3.7 K was measured when driven supercritically at 40 kV cm$^{-1}$ (Supplementary Fig. 15). This represents a temperature change increase of 60% compared with previous studies done on a less ordered PST[12–15] (cf Table 1 and Supplementary Fig. 7). As shown in Fig. 3a, the adiabatic temperature changes at field on ($\Delta T_{adiab,on}$) and at field off ($\Delta T_{adiab,off}$) do not peak at the same starting temperature $T_s$. This is due to the reversibility of the EC effect. Indeed, the temperature difference between the peaks is equal to the adiabatic temperature change. This mechanism is explained in[21], in which they show that if an EC process is reversible, the relation $\Delta T_{adiab,on}$ ($T_s$, $E$) = $-\Delta T_{adiab,off}$ ($T_s$ + $\Delta T_{adiab,on}$ ($T_s$, $E$), $E$) must be verified (with $E$ applied electric field). This is exactly what we observed in bulk PST, as seen in Supplementary Fig. 13 confirming its reversibility.

In this study, we consider that $C_p \Delta T_{adiab}$ stands for a good approximation of the exchangeable heat $Q$ (Fig. 3a, right $Y$-axis). This assumption is often used in the literature[2,22]. However, bulk PST transition is strongly 1st order and therefore exhibits a latent heat, which could induce some variations in our estimation.

**Table 1 Influence of B-site cation order on EC performance and temperature window of PST bulk ceramics.**

| Material | $\Omega$ | $T_0$ (K) | $\Delta T_{adiab}$ (K) | Temp. range (K) | $E$ (kV cm$^{-1}$) | Ref. |
|---|---|---|---|---|---|---|
| PST sample 1 | 0.98 | 300 | 3.7 | | 40 | This work |
| PST sample 1 | 0.98 | 300 | 3.1 | 8.4 | 22 | This work |
| PST sample 1 | 0.98 | 300 | 2.6 | 2.5 | 11 | This work |
| PST sample 2 | 0.89 | 298 | 2.4 | 2.5 | 11 | This work |
| PST | 0.85 | 295 | 2.3 | | 50 | 13 |
| PST | 0.85 | 295 | 1.6 | 7 | 25 | 12 |
| PST | 0.80 | 295 | 2.2 | 14 | 26 | 14 |
| PST | 0.34 | 263 | 0.5 | 19 | 25 | 12 |
| PST | 0 | 263 | 0.18 | 40 | 25 | 12 |

For different B-site cation order $\Omega$, we compare the directly measured $\Delta T_{adiab}$ and temperature range (full width at 80 percent of the maximum $\Delta T_{adiab}$) of our PST samples with values from the literature. Data of $\Delta T_{adiab}$ of our PST sample 2 are presented in Supplementary Fig. 6.

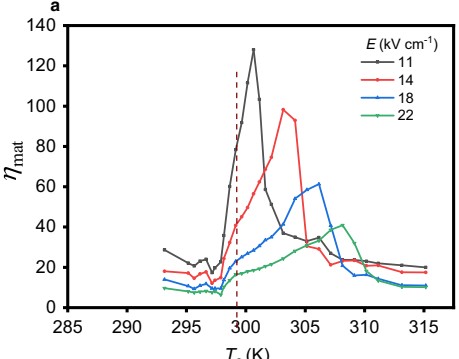

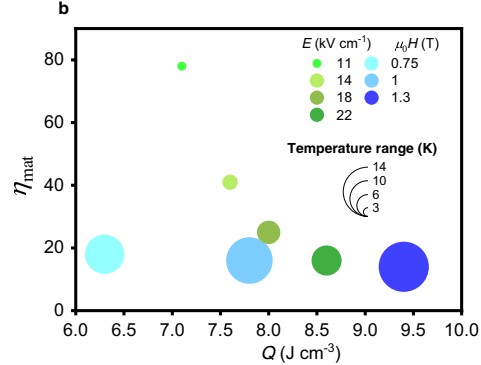

**Fig. 4 Materials efficiency $\eta_{mat}$ of PST. a** $\eta_{mat}$ of our bulk ceramic PST as a function of starting temperatures $T_s$ at different electric fields $E$. The dashed line shows the transition temperature where the material exchanges the most EC heat. $\eta_{mat}$ determined from $Q$ (Fig. 3a) and $W_e$ (Fig. 3c) measurements. **b** The bubble map represents $\eta_{mat}$ of PST (green bubbles) and Gd (blue bubbles) as a function of maximum caloric heat $Q$ per cm$^3$. The values of $Q$ are taken at the material transition temperature. The bubbles' color yields the field applied. The bubbles' diameter shows the temperature range of the material at a given field.

Consequently, we experimentally compared $C_p \Delta T_{adiab}$ and $-T_s \Delta S_{isothermal}$. Here $C_p$ is considered as constant and obtained from DSC (background value in Fig. 1b), $\Delta T_{adiab}$ is measured with the IR camera in adiabatic conditions and $\Delta S_{isothermal}$ comes from DSC measurements in which the electric field has been applied very slowly (200 s) to ensure isothermal conditions (Supplementary Fig. 23a). At the lowest values of electric fields (11 and 14 kV cm$^{-1}$), which are the most sensitive ones because the transition is not fully driven, both values of heat are similar (<5% apart) (Supplementary Fig. 23b). This justifies the assumption of using $C_p \Delta T_{adiab}$ as a good estimation of $Q$ in PST. Note also that considering the background value of $C_p$ constitutes a lower bound of $Q$, meaning that $\eta_{mat}$ cannot be overestimated with our method.

**Electrical work in PST.** Using a dedicated setup (Methods section), our bulk PST capacitors are charged at constant current ($I = 0.2$ mA) up to a given maximum voltage. Voltage is monitored in order to extract the electrical work $W_e = I \int U(t) \, dt$. These measurements are performed simultaneously with $\Delta T_{adiab}$. As seen in Fig. 3b, the voltage curves versus time show a change in slope from 295 to 304 K, which corresponds to the field-induced PE–FE phase transition. At 288 K (FE phase) and 313 K (PE phase), the capacitor is charged nearly linearly, indicating no phase change induced by the electric field. $W_e$ displayed in Fig. 3c peaks at 297 K at all fields. This is where PST requires the largest amount of energy to be charged. Finally, it is worth noting that the plateau observed in $Q$ at $T_0$ (Fig. 3a) combined with the sharp

peak in $W_e$ (Fig. 3c) suggests that $\eta_{mat}$ should display an optimum.

**PST materials efficiency.** Figure 4a shows $\eta_{mat}$ of PST versus $T_s$ at different values of the applied electric field. A maximum is observed at each value of electric field, sharper and higher at lower field. $\eta_{mat}$ reaches at maximum 128 at 11 kV cm$^{-1}$. This peak is very sharp on a temperature range of 2.5 K. At 22 kV cm$^{-1}$, $\eta_{mat}$ decreases down to 40, though in a wider temperature range of 8.4 K. This large variation in $\eta_{mat}$ is explained by the combination of $W_e$ peaking at 297 K for all fields and $Q$ experiencing a plateau which breadth increases with field. The peak moves towards higher temperature with field. Interestingly, the maximum in $\eta_{mat}$ is not obtained when $Q$ is maximum (dashed line in Fig. 4a).

Figure 4b displays the $\eta_{mat}$ of PST at different electric fields (green bubbles). Each point of $\eta_{mat}$ is taken when $Q$ is at maximum. At 22 kV cm$^{-1}$, $Q$ reaches 8.6 J cm$^{-3}$ and PST has an $\eta_{mat}$ of 16 over a temperature range of 8.4 K. However, by applying half this field (11 kV cm$^{-1}$), $Q$ remains substantial (7.1 J cm$^{-3}$) and $\eta_{mat}$ is multiplied by four, reaching 78. This fourfold increase is however associated with a threefold decrease of its temperature range as presented in Fig. 4b. These large values accompanied with these abrupt variations are a consequence of the strong first-order behavior of highly ordered PST.

Finally, in Table 2 and Fig. 4b, we compare PST's $\eta_{mat}$ with $\eta_{mat}$ of the MC material Gd (Supplementary Note 7) calculated in

**Table 2 Comparison of materials efficiency of PST with Gadolinium.**

|  | Bulk ceramic PST (sample 1) | | Gadolinium | |
| --- | --- | --- | --- | --- |
| Field | 11 kV cm$^{-1}$ | 22 kV cm$^{-1}$ | 1 T | 1.3 T |
| $Q$ (J cm$^{-3}$) | 7.1 | 8.6 | 7 | 9.4 |
| Temperature range (K) | 2.5 | 8.4 | 14 | |
| $\eta_{mat}$ | 78 | 16 | 18 | 14 |

We compare the materials efficiency of PST with Gd for similar heat exchanged $Q$.

an equivalent way. Although Gd exhibits a second-order phase transition contrary to PST, it is of interest to compare their materials efficiency because they are both considered the benchmark materials of their respective caloric family[3]. Hence, we computed direct values of $\eta_{mat}$ of Gd from the data in Bjork et al.[23] using the formalism of Heine[24], already used by Moya et al.[8] to extract $\eta_{mat}$ of MC materials. In[23], Gd was exposed to the magnetic field of permanent magnets of various intensity, namely 0.75, 1 and 1.3 T (cf Supplementary Note 7). As seen in Fig. 4b, PST can typically experience similar values of $Q$ as Gd, such as 8.6 J cm$^{-3}$ at 22 kV cm$^{-1}$ and 1 T, respectively. Note that Gd can easily survive larger values of magnetic field whereas it is not the case of bulk PST for which values beyond 22 kV cm$^{-1}$ applied for a long time are not safe. A striking point is that both PST and Gd exhibit a linear decrease of $\eta_{mat}$ versus $Q$, but with very different slopes. Indeed, $\eta_{mat}$ in PST varies between 78 and 20 over the entire range of $Q$ whereas Gd remains around 20. The much larger $\eta_{mat}$ in PST is nonetheless associated with a narrower temperature range, as little as 2.5 K, where $\eta_{mat}$ is maximum. On the contrary, the temperature range in Gd remains large and stable (between 10 and 14 K) as is typical of materials with a second-order phase transition.

## Discussion
The large values of $\eta_{mat}$ in bulk PST are one order of magnitude larger than those previously reported for EC materials, exclusively in thin films[7–9] and MLCs[20]. This places bulk PST among the best caloric materials in terms of materials efficiency (Supplementary Fig. 16). This is even without considering extrinsic means to recover energy, which promises even higher efficiency[10]. Besides, the comparison of PST with Gd shows that the first-order character of PST substantially increases $\eta_{mat}$ at the expense of working in a narrow temperature range. On the contrary, Gd has a lower $\eta_{mat}$ but enables a larger temperature range, which can be ascribed to its second-order smoother phase transition. It is also worth noting that keeping a high degree of order $\Omega$ in PST ensures a strong first-order phase transition in PST[25], and thus larger values of $Q$ and $\eta_{mat}$.

Hence, the giant material efficiency observed in bulk PST stems from its first-order nature. A stringent associated constraint is that it occurs at well-defined temperatures. In the case of a realistic EC heat pump, it is legitimate to wonder how one could run a useful cycle and what the impact on $\eta_{mat}$ would be. A first possibility is to use it in a "one-shot" mode that is typically useful in electronic apparatus with a temperature threshold, as described in[26]. In this case, the maximum efficiency would be reached at the optimal temperature, meaning at 301 K according to Fig. 3a at 11 kV cm$^{-1}$, which is functional but quite limiting. A more suitable use is to integrate this material into a regenerator. The interest of such a device is to build a gradient of temperature larger than $\Delta T_{adiab}$ by running a specific thermodynamic cycle[4]. To do so, the positive $\Delta T_{adiab}$ must be triggered at a temperature higher than when the negative $\Delta T_{adiab}$ is triggered in this cycle. Otherwise,

the built-in gradient would be destroyed. Moreover, we know that in the steady state positive and negative $\Delta T_{adiab}$s must exhibit the same magnitude. We experimentally mimicked the working principle of such a device on our samples with a temperature-controlled hot plate stage, as depicted in Supplementary Note 14. The idea is to build symmetrical cycles in the $\Delta T = f(T)$ map (Fig. 3a and Supplementary Fig. 20c). For instance, at 11 kV cm$^{-1}$, it is possible to obtain identical positive and negative $\left| \Delta T_{adiab} \right| = 1.7$ K if we trigger them respectively at 300.5 and 300 K (Regenerator A, Supplementary Fig. 20a). At such values, we measured $\eta_{mat}$ of at least 92. The price to pay is that $Q$ decreases (4.63 J cm$^{-3}$). Other functional examples are given in Supplementary Note 14. Moreover, we checked that bulk PST can be run in a regenerator by simulating it with a proven finite element model, detailed in[4] (Supplementary Fig. 21). Hence, although the constraints imposed by PST first-order nature are rather strong, it is possible to find experimental conditions enabling taking advantage of PST bulk giant $\eta_{mat}$.

Two solutions can be envisioned to benefit from large efficiencies of PST in future EC cooling prototypes and overcome its narrow temperature window. The first one is the use of PST in the forms of Multi-Layers Capacitors (MLCs). Nair et al.[20] showed that PST MLCs exhibit large $\Delta T_{adiab}$ over a temperature range of more than 73 K. However, $\eta_{mat}$ in PST MLCs decreases down to 7, probably because of a behavior closer to a second-order phase transition induced by the large field driving PST in a supercritical regime[20]. Indeed, we showed that our bulk PST sample exhibits a breakdown field close to 40 kV cm$^{-1}$ whereas PST MLCs can survive fields as large as 290 kV cm$^{-1}$. The second solution is based on multiple PST materials with different transition temperature. These graded PST ceramics could then be assembled in a layered regenerator structure, suggested by Olsen in the early eighties[27]. This solution has already been applied in several MC regenerators as for instance with La(Fe, Si, Mn)$_{13}$H$_y$, a sharply first-order phase transition material[28]. Each piece of material in the active regenerator has to operate close to its respective transition temperature in order to obtain a large temperature span in the device. A way to tune the transition temperature in EC materials is doping[2]. Shebanov[12] showed that dopants such as Sb, Y, Co, Nb, and Ti shift the transition temperature in PST from −7 K to +19 K.

Our study has shown the EC properties of highly ordered bulk ceramic lead scandium tantalate. We observed that the large degree of order in bulk PST enables increasing the maximum adiabatic temperature variation up to 3.7 K. Besides, bulk PST reaches materials efficiency values up to 128 thanks to its strong first-order phase transition, placing highly ordered bulk PST among the best caloric materials in terms of materials efficiency. Hence, bulk PST can be four times more efficient than the magnetocaloric material Gd at similar caloric heat exchanged, though in a narrow temperature range. In future, bulk PST might be considered as an interesting material for heat pumps based on layered regenerators. This should allow reaching very large coefficients of performances.

## Methods
**PST fabrication.** Oxide powders of PbO (99.9%, Sigma Aldrich), Sc$_2$O$_3$ (99.99%, Sigma Aldrich), and Ta$_2$O$_3$ (99.9%, Kojundo Chemical) were mixed according to the stoichiometric formula of Pb(Sc$_{1/2}$Ta$_{1/2}$)O$_3$, followed by 24 h of ball milling in ethanol using zirconia grinding media. The ball-milled powder was calcined at 850 °C for 2 h, and then crushed and ball-milled again for another 24 h. The powders were pelletized into disks of 10 mm in diameter under uniaxial press of 180 MPa with the addition of polyvinyl alcohol (PVA) as a binder. Sintering was conducted in a closed crucible with a mixture of PbZrO$_3$ and PbO as a sacrificial powder at 1300 °C for 2 h. The intended B-site ordering was induced by annealing at 1000 °C for 30 h.

**Infrared camera**. Measurements of $\Delta T_{\text{adiab}}$ were done using an IR Camera (X6580sc FLIR) on a 0.5 mm-thick ceramic PST capacitor. Sample 1 was completely covered with silver paste on its bottom side and on 0.480 cm$^2$ of its top side to avoid any short circuit. To get accurate measurements of the sample temperature with the IR camera, the top side of the sample was painted black to increase its emissivity to 1. $\Delta T_{\text{adiab}}$ data were collected on fast application and removal of an electric field on the electroded 0.480 cm$^2$ area. The sample was placed on a hot plate in a Linkam cell to control its starting temperature. The material was first cooled in the absence of electric field to 283 K and then heated to the desired temperature where three runs of electric field were applied. Therefore, three values of $\Delta T_{\text{adiab}}$ when applying the field and three values $\Delta T_{\text{adiab}}$ when removing the off field are measured. Measurements were carried out from 283 to 323 K every 0.5 K. At the fields and starting temperatures measured, the same $\Delta T_{\text{adiab}}$ was obtained on three electric field runs (Supplementary Fig. 11). A Keithley 2410 was used to apply voltage at constant current in the PST capacitor. A constant current of 0.2 mA was applied to guarantee adiabatic conditions with the surroundings. An automatized system that controls simultaneously the IR Camera, Linkam stage, and Keithley was done using Python to enable multiple and fast data acquisition. As the reproducibility of our measurements was checked after three cycles, each point of $\Delta T_{\text{adiab}}$ corresponds to an average of the three cycles (Supplementary Note 9).

**Dielectric measurements**. An impedance spectrometer was used to record dielectric measurements at 100 Hz on cooling and heating at a heating rate of 0.1 K min$^{-1}$ in a cryogenic probe.

**X-ray diffraction measurements**. XRD diffraction was carried out with PANalytical X'Pert Pro on the powdered PST from 15 to 60-degrees angle at every 0.02 degrees. The B-site cation order $\Omega$ was computed from the integrated intensities of the XRD maxima 111 and 200 (Supplementary Fig. 1) following the standard technique detailed in[11] and[29].

**Calorimetric measurements**. Using a commercial Differential Scanning Calorimeter (DSC), Mettler Toledo 3+, heat flow $dQ/dT$ measurements were done on 25.05 mg of bulk ceramic PST at a heating rate of 10 K min$^{-1}$. After removal of the baseline, the peaks of $dQ/dT$ were integrated (trapezoid method) to obtain latent heat $Q_0$ on cooling ($Q_{0,\text{c}} = \int_{T_{c1}}^{T_{c2}} \frac{dQ}{dT} dT$) and heating ($Q_{0,\text{h}} = \int_{T_{h1}}^{T_{h2}} \frac{dQ}{dT} dT$). On heating, $T_{h1}$ and $T_{h2}$ are chosen respectively before and after the transition and, on cooling, $T_{c1}$ and $T_{c2}$ are chosen respectively after and before the transition (Supplementary Note 2). The full entropy change across the transition $\Delta S_0$ at zero-field was calculated by dividing the latent heat $Q_0$ by the transition temperature $T_0$. Using sapphire as a reference with almost the same mass (25.5 mg) as PST, the zero-field specific heat of PST was determined using the formula $C_{\text{pPST}} = \frac{(dQ/dt)_{\text{PST}} \cdot m_{\text{sapphire}}}{m_{\text{PST}} \cdot (dQ/dt)_{\text{sapphire}}} C_{\text{Psapphire}}$. Masses were measured using a microbalance Mettler Toledo. $S'(T)$ which represents the entropy $S(T)$ referenced to the entropy at 280 K, far below $T_0$, was evaluated from calorimetric data using $S'(T, E) - S(T = 280\,\text{K}, E) = \int_{280\,\text{K}}^{T} \frac{dQ(T', E)/dT'}{T'} dT'$.

Isofield measurements were carried out at 10 K min$^{-1}$ on a customized DSC from Mettler Toledo. Constant electric field was applied from 283 to 323 K to measure the heat flow $dQ/dT$ under electric field. A quasi-direct measurements of entropy change $\Delta S$ driven by electric field was obtained using $\triangle S(T, E) = S'(T, E) - S'(T, 0)$ with $S'$ the entropy curve at a given electric field referenced at 280 K.

**Dedicated setup**. A code script written in Python was used to collect data of voltage and current as a function of time during the charge and discharge of the PST capacitors. The capacitors were charged at a constant current of 0.2 mA for less than 0.1 s, up to a target voltage. The current was chosen high enough to guarantee good adiabaticity in the IR camera measurements and small enough to be able to acquire voltage and current data as a function of time with a sufficient number of points (Supplementary Fig. 10b).

**Joule heating assessment**. We measured a stable leakage current $I_{\text{leak}} = 0.2\,\mu\text{A}$ when maximum voltage (1100 V) was applied during the thermalization phase of bulk PST sample 1. This induces a Joule heating power of 0.2 mW, which represents less than 30 μJ of Joule heating during the 0.12 s of the adiabatic step. In Fig. 3c, $W_{\text{e}}$ is always beyond 2 mJ when 1100 V have been applied (corresponding to 22 kV cm$^{-1}$ in our sample). Therefore, the influence of Joule heating in $W_{\text{e}}$ is <1.5%, which can be considered as negligible. Besides, we could not observe any influence of Joule heating in the IR camera characterization. In Supplementary Fig. 10a, the stable temperature after thermalization following the application or the removal of the electric field is the same in both cases. We also display in Supplementary Fig. 14 the IR images of PST sample 1 during a standard IR sequence, namely stable initial temperature, adiabatic heating step (field on), thermalization (field on), cooling step (field off), thermalization (field off), in which the colors displayed by thermalized steps—representative of sample's temperature—are the same as the initial step. This means that Joule heating also plays a

negligible role in the determination of $\Delta T_{\text{adiab}}$ — and therefore of heat — in our experiments.

## Data availability
All relevant data are available on request from the corresponding authors.

## Code availability
Computational codes for data collection, processing and analysis are available from the corresponding author on request.

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

## Acknowledgements

The authors want to thank N.D. Mathur, X. Moya, and B. Nair for fruitful discussions. This study has been partly funded by Fonds National de la Recherche (FNR) Luxembourg through the projects CAMELHEAT C17/MS/11703691/Defay and MASSENA PRIDE/15/10935404/Defay-Siebentritt.

## Author contributions

Y.N. ran the experiments with the help of P.L. and A.T. R.F. ran experiments at the early stage of the study. The fabrication of the PST bulk ceramics was realized by C.H.H. and W.J. Y.N. and E.D. wrote the paper with contributions from W.J. C.R.H.B. provided the data on Gd, contributed to set the experimental methodology and reviewed the original draft. E.D. conceived the idea, acquired the funding, and supervised the project.

## Competing interests

The author declares no competing interests.
