## [Peer Review File · Nature Communications]

REVIEWER COMMENTS

Reviewer #1 (Remarks to the Author):

The recent works reported that Lead Scandium Tantalate (PST) multi-layer capacitors show great potential in electrocaloric cooling. In the manuscript entitled "An energy efficient electrocaloric material", Nouchokgwe et al. studied the effects of B-site cation order on the electrocaloric performance and temperature window of PST bulk ceramics, as well as its materials efficiency. Based on the direct measurements of exchangeable heat and electrical work of PST bulk ceramics during electrocaloric effect, the materials efficiency defined as the ratio of the exchangeable electrocaloric heat to the work needed to trigger this heat was calculated. Comparing with other electrocaloric materials and magnetocaloric material Gd, the manuscript found that the large degree of order can increase the maximum adiabatic temperature change and materials efficiency, especially the highly ordered PST bulk ceramics possessing an excellent materials efficiency. The results are interesting and relevant to possible cooling applications. Some issues outlined below need to be taken into full consideration before the manuscript can be suitable for its publication.

1. In the manuscript, the temperature change is considered as adiabatic temperature change, while the heat transfer between the sample and surroundings were observed in Supplementary Fig. 9.1 and Supplementary Fig. 10. Does it suggest that the sample would be not under a adiabatic condition? Please explain this.

2. Around the phase transition temperature of PST, the absolute values of adiabatic temperature changes show a large difference between the case at field on and the case at field off, as shown in Supplementary Fig. 9.1(a) and other figures. Generally, the absolute values of adiabatic temperature changes for the both cases show only a very small difference or equal to each other. The authors are encouraged to explain its mechanism. In addition, one cycle cooling includes both the application and removal of electric field, the effects of this asymmetry adiabatic temperature change on the material efficiency or device's Coefficient Of Performance are encouraged to discuss in the manuscript.

3. In supplementary Fig. 9.2, the peak of adiabatic temperature change is shifted to higher temperature for the case at field off, while it is shifted to lower temperature for the case at field on. Please explain why.

4. In the sentence "IR camera measurements of reversible ΔT_{adiab}". As the authors claimed, PST shows a first-order phase transition, and latent heat should accompany first-order phase transitions. In addition, such transition would be associated with a part of irreversibility heat and, therefore, this irreversibility heat cannot be used in refrigeration. Can they please comment on the effect of such a latent heat upon application, and removal of the electric field, and its effects on material efficiency and device's Coefficient Of Performance?

5. As claimed in the sentence "Q is calculated from directly measured and constant background value of C_p ." In the manuscript, some calculations are done using constant background value of C_p , while the experiment data indicated that C_p is sensitive to temperature and external electric field. Can the authors comment on the effects of using constant background value of C_p in the calculations?

6. On page 16, "referenced to the entropy at 270 K, far below" and "with S' the entropy curve at a given electric field referenced at 280 K.". Different temperatures (270 K and 280 K) were chosen, please explain this.

7. Please double check some descriptions in the manuscript and Supplementary Information. For example,

the vertical coordinate in the lower plot of Fig. 1a.

"From specific heat measurements (Supplementary Fig.4)".

On page 8 line151, ".....(2), where ρ is the density"

8. For the convenience of readers, some details should be added. For example, how to determine the values of T_{c1} , T_{c2} , T_{h1} , and T_{h2} ?

Adiabatic temperature change of PST bulk ceramics were measured by the authors, and adiabatic

condition is difficult to achieve, how to achieve and control the adiabatic space and condition?

9. The authors are suggested to use only one symbol (K) to represent the unit of temperature throughout the whole manuscript.

10. Full description should be provided when its simplified form appears for the first time. For example, PST.

Reviewer #2 (Remarks to the Author):

Authors reported the huge energy efficiency of bulk PST. The value itself is quite significant for EC research field. Basically, dielectrics with less loss are very energy efficient materials.

Though EC has been studied actively since early 2000, real commercial cooling device has not been reported yet. In addition, large cooling device can not compete with mechanical cooling device. I think EC will be useful for cooling the small electronic device. Then the target temperature may be around 70°C. As many PST material properties are known till now, a little mention on device structure will be very useful to attract the attention.

PST has T_c near room temperature. What is the potential application of this material? If authors describe potential application, then this manuscript will attract more attention from many readers.

As authors know well and described in manuscript, there are many reports on MLC and thin film PST. I think the reason that MLC and thin film are used is to lower the driving voltage. Driving voltage is quite important for real applications.

What is the sample thickness and it is highly recommended to mention the real voltage rather than E-field. Ceramic is quite weak for strong E-field. If the P-E behaviour with the information on breakdown field is provided, then this information is useful to expand this material to be utilized for EC devices.

Reviewer #3 (Remarks to the Author):

The paper "An energy efficient electrocaloric material" by Nouchokgwe et al. is an interesting paper on the demonstration of a giant electrocaloric material's energy efficiency (η_{mat}) in a highly ordered bulk Lead Scandium Tantalate (PST) ceramic. The experimental data are of high quality and also well backed-up by direct measurements of heat and electrical work which is needed for the calculation of the electrocaloric figure of merit (η_{mat}). Further, the authors have successfully compared the energy efficiency of PST with respect to the benchmark magnetocaloric material Gd which highlights the efficacy of PST as a candidate electrocaloric material. The manuscript is in general of high quality, well written with new results that can contribute to new studies within the community. My final decision is to publish this paper following some minor technical comments:

a. In my opinion, the title of the paper is a bit obscure since there are several approaches to calculate the energy efficiencies of electrocaloric materials and the authors have used only one such approach in this paper. Further the name of the material in the title would provide the readers of the materials research community a better idea of the paper. I recommend changing the title...may be something like "Giant electrocaloric material's energy efficiency in highly ordered PST".

b. While calculating the values for the isothermal heat Q from the directly measured adiabatic temperature changes (Fig. 3 a) and also from the isothermal entropy changes (Fig. 2b), the authors consider a constant specific heat (C_p) independent of field and temperature. The authors need to justify this assumption since it is clearly visible that C_p is both temperature and field dependent for the PST sample (Fig. 1b and Supplementary Fig. 3d).

c. PST exhibits a first-order phase transition which is accompanied by hysteresis losses. The authors need to comment on how the hysteresis losses in PST affect the materials energy efficiency (η_{mat}). This is critical when one compares the material's efficiency (η_{mat}) with Gd which exhibits a second-order phase transition with no hysteresis losses.

d. During the IR camera measurements of reversible adiabatic temperature change, the effect of Joule heating due to leakage currents in the sample needs to be carefully taken into consideration. The authors mention that "This was done without Joule heating as the initial temperature of the sample was recovered after application of maximum field 22 kV cm^{-1} ." However with sufficient time the initial temperature of the sample is likely to be recovered so the statement needs to be revised. I recommend providing some of the actual IR images of the PST sample during field cycling in the Supplementary Information. Further leakage currents should be measured to make sure that the direct measurements of the adiabatic temperature changes by IR imaging are not affected by extrinsic factors.

e. Table 1 and table 2 are not separately provided. This needs to be corrected.

f. The authors claim that the high efficiency observed in the PST sample is due to its high degree of ordering. However, the authors use only powder XRD for calculating the degree of ordering in PST which could have some errors. It will be beneficial to the readers if the authors can cite some references in the text in this regard.

Answers to the reviewers

Italic – comments from reviewers

Standard text – answers to the reviewers

Highlighted yellow – modifications implemented in the text

Reviewer #1 (Remarks to the Author):

The recent works reported that Lead Scandium Tantalate (PST) multi-layer capacitors show great potential in electrocaloric cooling. In the manuscript entitled "An energy efficient electrocaloric material", Nouchokgwe et al. studied the effects of B-site cation order on the electrocaloric performance and temperature window of PST bulk ceramics, as well as its materials efficiency. Based on the direct measurements of exchangeable heat and electrical work of PST bulk ceramics during electrocaloric effect, the materials efficiency defined as the ratio of the exchangeable electrocaloric heat to the work needed to trigger this heat was calculated. Comparing with other electrocaloric materials and magnetocaloric material Gd, the manuscript found that the large degree of order can increase the maximum adiabatic temperature change and materials efficiency, especially the highly ordered PST bulk ceramics possessing an excellent materials efficiency. The results are interesting and relevant to possible cooling applications. Some issues outlined below need to be taken into full consideration before the manuscript can be suitable for its publication.

1. In the manuscript, the temperature change is considered as adiabatic temperature change, while the heat transfer between the sample and surroundings were observed in Supplementary Fig. 9.1 and Supplementary Fig. 10. Does it suggest that the sample would be not under an adiabatic condition? Please explain this.

Answer: We thank the referee for this comment on the adiabaticity conditions of our measurements.

ΔT_{adiab} on field and ΔT_{adiab} off field correspond respectively to the jump in temperature in step 2 and step 4 in Supplementary Fig. 9.1. The adiabaticity of the measurements is controlled by the current applied in our PST capacitor.

Our adiabatic temperature change measurements ΔT_{adiab} were collected under adiabatic conditions as for currents ≥ 0.2 mA, the peak of ΔT_{adiab} doesn't change with increasing current (figure below). Besides at 0.2 mA, the application/removal time of the electric field is about 30 times faster than the relaxation time. However, for current smaller than 0.2 mA, the peak of ΔT_{adiab} increases with the current. This area is considered as non-adiabatic as the sample exchanges with surroundings.

Modification:

In the text

In the 'Electrocaloric exchangeable heat in PST' section, we added a note regarding adiabatic conditions in Supplementary.

Under the fast (less than 0.1 s) application and removal of an electric field at the same starting temperature, positive and negative ΔT_{adiab} respectively, were measured on three successive electric field cycles using the protocol described in the Methods Section (see also Supp. Fig. 9.3 about adiabatic conditions).

In Supplementary Information

In Supplementary Information 9, the paragraph below on adiabatic conditions was added along with the figure below (Supp. Fig. 9.3).

Adiabatic conditions: The adiabaticity of the measurements is controlled by the current applied in our PST capacitor. The charging time of the sample must be much faster than the characteristic time of thermal relaxation to guarantee adiabaticity. As presented in Supp. Fig 9.3, for current lower than 0.2 mA, the thermal conditions are non-adiabatic because ΔT is lower than the asymptotic value of ΔT_{adiab} . When current ≥ 0.2 mA, ΔT_{adiab} stays constant and maximum, meaning that the expected adiabatic conditions are fulfilled. At 0.2 mA, the application/removal time of the electric field takes 0.12 s, which is about 30 times faster than the thermal relaxation time experienced by PST in our experiments (see Fig. S9.1a).

Supplementary Fig. 9.3 | **Adiabatic conditions.** $\Delta T_{adiab,on}$ versus applied current. The measurements were done on PST sample 1 bulk at 22 kV cm^{-1} .

2. a. Around the phase transition temperature of PST, the absolute values of adiabatic temperature changes show a large difference between the case at field on and the case at field off, as shown in Supplementary Fig. 9.1(a) and other figures. Generally, the absolute values of adiabatic temperature changes for the both cases show only a very small difference or equal to each other. The authors are encouraged to explain its mechanism.

Answer: From the adiabatic temperature measurements ΔT_{adiab} , one can indeed see that at some starting temperatures T_s (around phase transition), ΔT_{adiab} field on is different in magnitude from ΔT_{adiab} field off. It is intrinsic to the material and due to the fact that the zero field and finite field of the entropy curves converges at low and high temperatures. The difference in magnitude is large due to the very sharp transition of our highly ordered PST. This mechanism is explained in [B. Nair, Ph.D. thesis, University of Cambridge, 2020, available at <https://www.repository.cam.ac.uk/handle/1810/312805>]. The figure below was taken from B.Nair PhD Thesis.

[Redacted]

This asymmetry in the adiabatic temperature was also observed in PST MLCs where the difference in magnitude is small due to the broad transition of PST MLCs [Nair, B. *et al. Nature* **575**, 468-472 (2019)]. Asymmetric in ΔT_{adiab} was also reported in a less ordered bulk ceramic PST [Stern-Taulats, E., PhD Thesis, *Universitat de Barcelona* (2017)] and in a 1st order transition magnetocaloric material (Fe-Rh) [Stern-Taulats, E. *et al., APL* **107**, 152409 (2015)].

Modification: A section “Asymmetry in adiabatic measurements” was added in Supplementary Information 9. The following paragraph has been added at the end of the latter.

The fact that ΔT_{adiab} field on is different in magnitude from ΔT_{adiab} field off is intrinsic to the material and due to the fact that the zero field and finite field entropy curves (Supp. Fig. 3.1) get closer at low and high temperatures [S8]. In the case of bulk PST, the difference in magnitude is large because of the very sharp first order transition. This asymmetry in the adiabatic temperature was also observed in PST MLCs [S8-S9] where the difference in magnitude is small due to the broad transition of PST MLCs. Asymmetric in ΔT_{adiab} was also reported in less ordered bulk ceramic PST [S10] and in a 1st order transition magnetocaloric material (Fe-Rh) [S11].

[S8] Nair, B., Ph.D. thesis, University of Cambridge, 2020, available at <https://www.repository.cam.ac.uk/handle/1810/312805>

[S9] Nair, B. *et al.* Large electrocaloric effects in oxide multilayer capacitors over a wide temperature range. *Nature* **575**, 468-472 (2019).

[S10] Stern-Taulats, E., PhD Thesis, *Universitat de Barcelona* (2017)

[S11] Stern-Taulats, E. *et al., APL* **107**, 152409 (2015)

2. b. In addition, one cycle cooling includes both the application and removal of electric field, the effects of this asymmetry adiabatic temperature change on the material efficiency or device's Coefficient Of Performance are encouraged to discuss in the manuscript.

Answer: We thank the reviewer for this important comment, giving us the opportunity to improve our manuscript.

Modifications:

- In the article: The paragraph below has been added in the section *Discussion* (second paragraph).

Hence, the giant material efficiency observed in bulk PST stems from its first-order nature. A stringent associated constraint is that it occurs at well-defined temperatures. In the case of a realistic EC heat pump, it is legitimate to wonder how one could run a useful cycle and what the impact on η_{mat} would be. A first possibility is to use it in a 'one-shot' mode that is typically useful in electronic apparatus with a temperature threshold, as described in [25]. In this case, the maximum efficiency would be reached at the optimal temperature, meaning at 301 K according to Fig. 3a at 11 kV cm^{-1} , which is functional but quite limiting. A more suitable use is to integrate this material into a regenerator. The interest of such a device is to build a gradient of temperature larger than ΔT_{adiab} by running a specific thermodynamic cycle [4]. To do so, the positive ΔT_{adiab} must be triggered at a temperature higher than when the negative ΔT_{adiab} is triggered in this cycle. Otherwise, the built-in gradient would be destroyed. Moreover, we know that in the steady state positive and negative ΔT_{adiab} must exhibit the same magnitude. We experimentally mimicked the working principle of such a device on our samples with a temperature-controlled hot plate stage, as depicted in Supp. Fig. 14. The idea is to build symmetrical cycles in the $\Delta T = f(T)$ map (Fig 3a and Supp. Fig. 14c). For instance, at 11 kV cm^{-1} , it is possible to obtain identical positive and negative $|\Delta T_{\text{adiab}}| = 1.7 \text{ K}$ if we trigger them respectively at 300.5 K and 300 K (Regenerator A, Supp. Fig. 14a). At such values, we measured η_{mat} of at least 92. The price to pay is that Q decreases (4.63 J cm^{-3}). Other functional examples are given in Supp. Info. 14. Moreover, we checked that bulk PST can be run in a regenerator by simulating it with a proven finite element model, detailed in [4] (Supp. Fig. 14.1). Hence, although the constraints imposed by PST first-order nature are rather strong, it is possible to find experimental conditions enabling taking advantage of PST bulk giant η_{mat} .

[4] Torello, A. *et al.* Giant temperature span in electrocaloric regenerator. *Science* **370**, 125-129 (2020).

[25] Defay, E., Mathur, N., Kar-Narayan, S., Soussi, J., Method for limiting the variation in the temperature of an electrical component, **US9326423B2**, 2012.

- In Supplementary: Simulations and experiments discussed in the previous paragraph have been added in Supplementary 14.

14 - Potential Application of bulk PST

We show below how our PST could be used in a fluid-based regenerator without being affected by the asymmetry of ΔT_{adiab} , irreversibility or hysteresis losses. From ΔT_{adiab} measurements (Supp. Fig.14c) one can define a window of temperature where the regenerator could operate. At the maximum materials efficiency obtained at 11 kV cm^{-1} , we could simulate experimentally reproducible and reversible regenerators (Supp. Fig.14a and 14b). The regenerator would operate in a small temperature span at a lower heat exchanged (Supp. Fig. 14d). The heat can be increased by increasing the applied electric field but the materials efficiency will decrease as the heat saturates while the electrical work increases (Supp. Fig 14.2).

Furthermore, using the ΔT_{adiab} measurements in Supp. Fig.14c, we did some simulations of a fluid-based regenerator based on a similar model published in [S13] operating at different starting temperatures around room temperature and without any kind of heat losses to the environment. We show that a temperature gradient can be reached in all cases after several cycles (Supp. Fig 14.1). This indicates that, despite an asymmetry in ΔT_{adiab} or hysteresis losses, bulk PST could be used to build different kinds of regenerator prototypes.

[S13] Torello, A. *et al.* Giant temperature span in electrocaloric regenerator. *Science* **370**, 125-129 (2020).

Supplementary Fig. 14 | Potential use of PST bulk in a regenerator. Here we show two potential working points of PST bulk if it were integrated in a regenerator. The cycle of the latter has been mimicked with a temperature-controlled hot plate stage (Linkam). In a (resp. b), PST is first set at 300.5 K (resp. 301 K). The EC positive $\Delta T_{\text{adiab}} = 1.7$ K (resp. 1.4 K) is then triggered by charging PST. Heat is exchanged and PST goes back to 300.5 K (resp. 301 K). If PST was in a regenerator, a fluid (for instance) would then be displaced and PST temperature would decrease. Here, we suppose that this temperature is 300 K (resp. 299.5 K). This value is chosen in order to obtain a symmetrical position in $\Delta T_{\text{adiab}} = f(T_s)$ displayed in c. This EC negative $\Delta T_{\text{adiab}} = -1.7$ K (resp. -1.4 K) is then triggered by discharging PST and PST exchanges heat until it reaches 300 K (299.5 K) again. And the cycle carries on as it would in a proper regenerator. **a)** Regenerator A operating between 300-300.5 K, on a temperature window of 0.5 K. **b)** Regenerator B operating on a temperature window of 1.5 K. **c)** Adiabatic temperature change of PST bulk at 11 kV cm^{-1} . The orange and black lines represent respectively the regenerators A and B. **d)** the table shows for each regenerator, its temperature window, the heat exchanged and the corresponding measured materials efficiency.

Supplementary Fig. 14.1 | **Regenerator modelling.** The time evolution of the hot side (red) and cold side (blue) of an active regenerator based on the EC effect from Supp. Fig.14c at starting temperatures (a) 297 K, b) 299 K, c) 301 K and d) 303 K. e) shows the dimensions of the regenerator simulated. The simulation consists of a finite element method (FEM) 2D representation of an active regenerator made with a single PST plate of 0.2 mm x 4 cm. No losses to the surroundings were considered. All the parameters of this model are detailed in [S13].

Supplementary Fig. 14.2| **Potential regenerator for PST bulk at higher heat Q.** The heat exchanged can be increased by increasing the electric field but this will decrease the materials efficiency as shown in **d**). **a**) regenerator C at the electric field of 14 kV cm⁻¹ **b**) regenerator D at 18 kV cm⁻¹ **c**) regenerator E at field of 22 kV cm⁻¹ **d**) for each regenerator, the heat exchanged, materials efficiency and adiabatic temperature change.

3. In supplementary Fig. 9.2, the peak of adiabatic temperature change is shifted to higher temperature for the case at field off, while it is shifted to lower temperature for the case at field on. Please explain why.

Answer: The adiabatic temperature changes at field on ($\Delta T_{adiab,on}$) and at field off ($\Delta T_{adiab,off}$) do not peak at the same starting temperature T_s , indeed. This is due to the reversibility of the electrocaloric effect. Indeed, the temperature difference between the peaks is equal to the adiabatic temperature change. This mechanism is explained in Nielsen, K. et al., *Physical Review B* **81**, 054423(1-5) (2010). They show that if a process is reversible, the relation below must be verified:

$$\Delta T_{adiab,on}(T_s, E) = -\Delta T_{adiab,off}(T_s + \Delta T_{on}(T_s, E), E)$$

with E the applied electric field. This is exactly what we observed in bulk PST, confirming its reversibility.

Modifications: A section *Asymmetry in adiabatic measurements* was added in Supplementary Information 9 explaining this mechanism along with the Supplementary Fig. 9.4. An explanation on asymmetry was added in the paper as well.

In Supplementary:

Asymmetry in adiabatic measurements

One can see in Supp. Fig. 9.2. and Fig. 3a that the adiabatic temperature changes at field on ($\Delta T_{adiab,on}$) and at field off ($\Delta T_{adiab,off}$) do not peak at the same starting temperature T_s . This is due to the reversibility of the electrocaloric effect. Indeed, the temperature difference between the peaks is equal to the adiabatic temperature change (see Supp. Fig. 9.4). This mechanism is explained in [S7], in which they show that if a process is reversible, the relationship below must be verified: $\Delta T_{adiab,on}(T_s, E) = -\Delta T_{adiab,off}(T_s + \Delta T_{on}(T_s, E), E)$, with E the applied electric field. This is exactly what we observed in bulk PST.

[S7] Nielsen, K. K., Bahl, C. R. H. & Smith, A. Constraints on the adiabatic temperature change in magnetocaloric materials. *Physical Review B* **81**, 054423(1-5) (2010).

Supplementary Fig. 9.4 | **Reversibility of EC measurements.** Here we show ΔT_{adiab} of PST bulk at 22 kV cm^{-1} versus starting temperature T_s . Red curves and dark blue curves are respectively ΔT_{adiab} due to the EC effect when the field is on and off and measured experimentally with an IR camera. The cyan dash curve is the ΔT_{adiab} due to EC effect when the field is off and derived from the equation above. $\Delta T_{adiab,off}$ **exp** and $\Delta T_{adiab,off}$ **derived** coincide well and thereby validate the reversibility of our EC measurements.

In the Article: The paragraph below has been added to the paper, in the section: Electrocaloric exchangeable heat in PST.

As shown in Fig. 3a (more details in Supp. Fig. 9.4), the adiabatic temperature changes at field on ($\Delta T_{adiab,on}$) and at field off ($\Delta T_{adiab,off}$) do not peak at the same starting temperature T_s . This is due to the reversibility of the electrocaloric effect. Indeed, the temperature difference between the

peaks is equal to the adiabatic temperature change. This mechanism is explained in [21], in which they show that if an EC process is reversible, the relation $\Delta T_{adiab,on}(T_s, E) = -\Delta T_{adiab,off}(T_s + \Delta T_{on}(T_s, E), E)$ must be verified (with E applied electric field). This is exactly what we observed in bulk PST, confirming its reversibility.

[21] Nielsen, K. et al., *Physical Review B* **81**, 054423(1-5) (2010)

4. In the sentence “IR camera measurements of reversible $\det T_{\{adiab\}}$”. As the authors claimed, PST shows a first-order phase transition, and latent heat should accompany first-order phase transitions. In addition, such transition would be associated with a part of irreversibility heat and, therefore, this irreversibility heat cannot be used in refrigeration. Can they please comment on the effect of such a latent heat upon application, and removal of the electric field, and its effects on material efficiency and device’s Coefficient Of Performance?

Answer: The answer to this question is addressed above in the answer of question 2b. We show how bulk PST could be used in a regenerator prototype without being affected by irreversible heat.

Moreover, we address the question of latent heat in the next question.

5. As claimed in the sentence “ Q is calculated from directly measured and constant background value of C_p .” In the manuscript, some calculations are done using constant background value of C_p , while the experiment data indicated that C_p is sensitive to temperature and external electric field. Can the authors comment on the effects of using constant background value of C_p in the calculations?

Answer: We thank the reviewer for this very important comment, given us the opportunity to improve our manuscript. To properly answer this question, we performed new experiments, namely isothermal entropy changes at different electric fields in DSC. This enabled us proving experimentally that $-T_s \Delta S_{isothermal}$ is very similar to $C_p \Delta T_{adiab}$ with C_p corresponding to the background value in our conditions.

Modifications

In the text (last paragraph in ‘Electrocaloric exchangeable heat in PST’

In this study, we consider that $C_p \Delta T_{adiab}$ stands for a good approximation of the exchangeable heat Q (Fig. 3a, right Y-axis). This assumption is often used in the literature [2, 29]. However, bulk PST transition is strongly 1st order and therefore exhibits a latent heat, which could induce some variations in our estimation. Consequently, we experimentally compared $C_p \Delta T_{adiab}$ and $-T_s \Delta S_{isothermal}$. Here C_p is considered as constant and obtained from DSC (background value in Fig 1b), ΔT_{adiab} is measured with the IR camera in adiabatic conditions and $\Delta S_{isothermal}$ comes from DSC measurements in which the electric field has been applied very slowly (200 s) to ensure isothermal conditions (Supp. Fig. 15a). At the lowest values of electric fields (11 and 14 kV cm⁻¹), which are the most sensitive ones because the transition is not fully driven, both values of heat are very similar (less than 5% apart) (Supp. Fig. 15b). This justifies the assumption of using $C_p \Delta T_{adiab}$ as a good estimation of Q in PST. Note also that considering the background value of C_p constitutes a lower bound of Q , meaning that η_{mat} cannot be overestimated with our method.

[29] S. G. Lu, B. Rožič, Q. M. Zhang, Z. Kutnjak, R. Pirc, Minren Lin, Xinyu Li, Lee Gornoy, Comparison of directly and indirectly measured electrocaloric effect in relaxor ferroelectric polymers, *Appl. Phys. Lett.* **97**, 202901 (2010)

Here is the text we suppressed from the initial version that has been replaced by the previous paragraph.

If one considers a constant C_p independent of field and temperature ($300 \text{ J K}^{-1} \text{ kg}^{-1}$), a $\Delta T \approx 3.3 \text{ K}$ can be deduced from $\Delta T \approx T \Delta S / C_p$, equation systematically used in the literature, [2–3]. Here ΔS comes from our DSC isofield values at 18 kV cm^{-1} and shown in Fig. 2c. This value is slightly higher than the one extracted from our IR measurements reaching 3.1 K (Fig. 3a). Consequently, in the following, we consider that ΔT_{adiab} enables obtaining a rather accurate estimation of the isothermal heat $Q = T \Delta S \approx \rho V C_p \Delta T_{\text{adiab}}$ (2), where ρ is the density and V is the volume (cf Fig. 3a). This allows the simultaneous measurement of heat and electrical work, and thus the deduction of fairly accurate values of η_{mat} versus applied field.

In Supplementary Information (1)

In Supp. Section 3, we added the following text and modified Supp. Fig. 3d.

The measurements of C_p under electric field are quite challenging (Supp. Fig 3d.) because it is difficult to obtain a flat baseline. Indeed, the latter is affected by the connecting wires attached with silver paste to the sample. The results displayed in Supp. Fig. 3d are representative of all the measurements performed under electric field. With the accuracy of our set-up, we could not deduce any variation of C_p baseline versus electric field. Consequently, C_p equals $300 \text{ J kg}^{-1} \text{ K}^{-1} \pm 20 \text{ J kg}^{-1} \text{ K}^{-1}$, which is in line with literature [S4, S8]. The clear influence of electric field is C_p peak shifting towards higher temperature.

[S4] Crossley, S., Nair, B., Whatmore, R. W., Moya, X. & Mathur, N. D. Electrocaloric cooling cycles in lead scandium tantalate with true regeneration via field variation. *Phys. Rev. X* **9**, 041002 (2019).

[S8] Nair, B., Ph.D. thesis, University of Cambridge, 2020, available at <https://www.repository.cam.ac.uk/handle/1810/312805>

Supplementary Fig.3| **Temperature, latent heat, entropy change and specific heat deduced from isofield Differential Scanning Calorimetry (DSC) measurements on PST sample 1** a) Transition temperature of PST versus electric field while heating $T_{0,h}$ and cooling $T_{0,c}$, b) latent heat Q_0 versus electric field, c) entropy change ΔS_0 versus electric field, d) Specific heat C_p measurements under electric field.

In Supplementary Information (2)

The Supplementary Figure 15 below has been added in Supplementary Information, supporting the paragraph added in the main text.

Electrocaloric Exchangeable heat

Supplementary Fig. 15 | Comparison between $C_p \Delta T_{\text{adiab}}$ and $-T_s \Delta S_{\text{isothermal}}$. a) Isothermal application of an electric field of 14 kV cm^{-1} using DSC. It is applied very slowly (200 s) to maintain the temperature almost constant and measure a DSC signal. The integral under the DSC signal corresponds to the isothermal heat exchange $-T_s \Delta S_{\text{isothermal}}$. b) comparison of $C_p \Delta T_{\text{adiab}}$ to $-T_s \Delta S_{\text{isothermal}}$ at two electric field values (11 and 14 kV cm^{-1}). ΔT_{adiab} is measured with the IR camera in adiabatic conditions and C_p is considered as constant ($300 \text{ J kg}^{-1} \text{ K}^{-1}$), which corresponds to the background value of C_p measurements (figure 1b). The good match between $C_p \Delta T_{\text{adiab}}$ to $-T_s \Delta S_{\text{isothermal}}$ proves that it is legitimate to consider $C_p \Delta T_{\text{adiab}}$ as an excellent estimation of the heat exchanged in η_{mat} .

6. On page 16, “referenced to the entropy at 270 K, far below” and “with S' the entropy curve at a given electric field referenced at 280 K.” Different temperatures (270 K and 280 K) were chosen, please explain this.

Thanks for the comment, this was a mistake.

Modifications: All the entropy curves at zero field and finite fields were all referenced at the same temperature **280 K**. Figure 1.c was corrected as well as its legend.

7. Please double check some descriptions in the manuscript and Supplementary Information. For example, the vertical coordinate in the lower plot of Fig. 1a. “From specific heat measurements (Supplementary Fig.4)”. On page 8 line151, “.....(2), where ρ is the density”

Thanks for correcting these typos. They were all corrected in the Supplementary and paper.

Modifications:

- In the paper, the vertical coordinate in the lower plot of Fig.1a. was corrected from $10^3 \tan \delta$ to $10^{-3} \tan \delta$.
 - In supplementary, “From specific heat measurements (Supplementary Fig.4)” was changed to “From specific heat measurements measured by differential scanning calorimetry (Supplementary Fig.3d)”
 - In the paper, on page 8 line151, “.....(2) where ρ the density” . We now have a new description of the exchangeable heat that directly impacts this comment, which is fully addressed in reviewer 1’s fifth comments.
-

8. For the convenience of readers, some details should be added. For example, how to determine the values of T_{c1} , T_{c2} , T_{h1} , and T_{h2} ?

Modifications: A sentence explaining how the values T_{c1} , T_{c2} , T_{h1} and T_{h2} were chosen was added in Supplementary Information 2.

T_{c1} , T_{c2} are respectively above and below the cooling peak. They are the extreme values for which heat flow dQ/dT is still naught before the latent heat peak starts appearing. T_{h1} and T_{h2} have been chosen similarly near the heating peak (cf Supplementary Fig. 2).

8.1 Adiabatic temperature change of PST bulk ceramics were measured by the authors, and adiabatic condition is difficult to achieve, how to achieve and control the adiabatic space and condition?

Answer: Thanks to the referee for his/her comment. We hope we addressed this question properly in reviewer 1's first comment.

9. *The authors are suggested to use only one symbol (K) to represent the unit of temperature throughout the whole manuscript.*

Answer: We thank for the reviewer for his/her suggestion.

Modification: Every value of temperature in degree Celsius ($^{\circ}\text{C}$) has been changed to Kelvin (K).

10. *Full description should be provided when its simplified form appears for the first time. For example, PST.*

Modifications: the modifications below were done in the article.

- *In abstract:* Lead Scandium Tantalate (PST)
- *In introduction:*
 - MC (magnetocaloric), EC (electrocaloric), and mC (mechanocaloric) materials
 - Gadolinium (Gd)
 - with a infrared camera (IR) camera
- *In section:* Electrocaloric effect in PST
 - X-Ray Diffraction (XRD)

Reviewer #2 (Remarks to the Author):

Authors reported the huge energy efficiency of bulk PST. The value itself is quite significant for EC research field. Basically, dielectrics with less loss are very energy efficient materials.

1. Though EC has been studied actively since early 2000, real commercial cooling device has not been reported yet. In addition, large cooling device can not compete with mechanical cooling device.

Answer We thank the reviewer for his comment. Indeed, no real commercial cooling device has been reported so far. However, three prototypes were recently reported, with two of them using PST as the working medium (cited in the introduction of the paper). This gives hope that possible real commercial cooling devices become true in the future. After discussions with five cooling industries, we learnt that the minimum temperature span required to eventually build a real commercial cooling device is 10 K. This threshold has been overcome in [Torello et al., Science, 2020] with a regenerator reaching a temperature span of 13 K.

More details are given in the next answers regarding applications.

2. As many PST material properties are known till now, a little mention on device structure will be very useful to attract the attention.

Answer: We thank the reviewer for his comment. Indeed, we have not provided a scheme of our device. A figure showing the structure of the PST capacitor was added in Supplementary Information 13.

Modification: The figure below was added as Supplementary Information 13.

Supplementary Fig. 13 | Structure of PST bulk material. The dielectric material PST is 0.5mm thick. The area of the top electrode is 0.480cm².

3. PST has T_c near room temperature. What is the potential application of this material? If authors describe potential application, then this manuscript will attract more attention from many readers.

Answer

The most natural application for EC materials is air conditioning, especially if it can be made energy efficient.

Modification (added in the introduction)

Lead Scandium Tantalate $\text{Pb}(\text{Sc}_{1/2}\text{Ta}_{1/2})\text{O}_3$ [11-16] is one of the most promising EC materials, already utilized to build EC heat pumps [4-5][17-19]. It exhibits a first-order ferroelectric to paraelectric phase transition near room temperature, which makes it attractive for air conditioning.

4. I think EC will be useful for cooling the small electronic device. Then the target temperature may be around 70°C.

Answer

Indeed, cooling electronic devices can be of interest, as already envisaged in one patent [25]. To do so with PST and take advantage of its large latent heat, dopants could be used to shift T_c (very end of the discussion).

[25] Defay, E., Mathur, N., Kar-Narayan, S., Soussi, J., Method for limiting the variation in the temperature of an electrical component, **US9326423B2**, 2012.

Modification

We mentioned the application about electronic devices with the associated patent in the discussion section in the main paper, together with the change of T_c brought about with dopants.

5. As authors know well and described in manuscript, there are many reports on MLC and thin film PST. I think the reason that MLC and thin film are used is to lower the driving voltage. Driving voltage is quite important for real applications.

What is the sample thickness and it is highly recommended to mention the real voltage rather than E-field. Ceramic is quite weak for strong E-field. If the P-E behaviour with the information on breakdown field is provided, then this information is useful to expand this material to be utilized for EC devices.

Answer:

We thank the reviewer for giving us the opportunity to clarify this point.

The sample is 0.5 mm thick (information given in the Method section), but we acknowledge that this information should be in the sample preparation section (PST fabrication) in the Method section. This is now the case.

Moreover, a table showing voltage with corresponding electric field was added in Supplementary Information 9. Voltages up to 1100 V were applied to the sample. It was difficult to apply higher voltages repeatedly due to short circuits with air and electrodes. Moreover, 1100V was enough to drive the full phase transition in PST, which limited the interest in increasing further voltage. However, as mentioned in the initial text, we could apply for a short duration voltage as high as 2000 V in the well-ordered 0.5 mm – thick PST sample, inducing ΔT_{adiab} as high as 3.7 K (supercritical regime). Consequently, we can say that the breakdown field of bulk PST is at least 40 kV cm^{-1} .

Besides, P(E) loops were carried out and added in Supplementary Information 12, as requested.

Modification

In the main text, ‘electrocaloric exchangeable heat in PST’ section

“IR camera measurements of reversible and reproducible ΔT_{adiab} performed on a 0.480 cm^2 area bulk ceramic **0.5 mm-thick PST sample 1**”

We also mentioned “**sample 1**” each time it was needed in the paper.

We added the following sentence about breakdown field in the discussion section.

“**Indeed, we showed that our bulk PST exhibit a breakdown field close to 40 kV cm^{-1} whereas PST MLCs can survive fields as large as 290 kV cm^{-1} .**”

In Supplementary

Supplementary Table 9 and supplementary Fig. 12

Voltage (V)	Electric Field (kV cm^{-1})
550	11
700	14
900	18
1100	22

Supplementary Table 9 | **Voltage – Electric field**. Here we give the real voltage applied in bulk ceramic PST sample 1 (0.5 mm-thick).

Supplementary Fig. 12 | **Polarisation-electric field loops of 0.5 mm-thick bulk PST sample 1 at different temperatures (from 288 K to 313 K)**. The measurements were done using a standard Sawyer-Tower circuit at 20 Hz.

Reviewer #3 (Remarks to the Author):

The paper "An energy efficient electrocaloric material" by Nouchokgwe et al. is an interesting paper on the demonstration of a giant electrocaloric material's energy efficiency (η_{mat}) in a highly ordered bulk Lead Scandium Tantalate (PST) ceramic. The experimental data are of high quality and also well backed-up by direct measurements of heat and electrical work which is needed for the calculation of the electrocaloric figure of merit (η_{mat}). Further, the authors have successfully compared the energy efficiency of PST with respect to the benchmark magnetocaloric material Gd which highlights the efficacy of PST as a candidate electrocaloric material. The manuscript is in general of high quality, well written with new results that can contribute to new studies within the community. My final decision is to publish this paper following some minor technical comments:

a. In my opinion, the title of the paper is a bit obscure since there are several approaches to calculate the energy efficiencies of electrocaloric materials and the authors have used only one such approach in this paper. Further the name of the material in the title would provide the readers of the materials research community a better idea of the paper. I recommend changing the title...may be something like "Giant electrocaloric material's energy efficiency in highly ordered PST".

Answer: We are thankful to the reviewer for suggesting this title which indeed gives a better idea of the paper.

Modification: we thank the reviewer for his suggestion. The title of the article has been changed and now it reads “Giant electrocaloric materials energy efficiency in highly ordered lead scandium tantalate”.

b. While calculating the values for the isothermal heat Q from the directly measured adiabatic temperature changes (Fig. 3 a) and also from the isothermal entropy changes (Fig. 2b), the authors consider a constant specific heat (C_p) independent of field and temperature. The authors need to justify this assumption since it is clearly visible that C_p is both temperature and field dependent for the PST sample (Fig. 1b and Supplementary Fig. 3d).

Answer: We thank the reviewer for this very important comment, given us the opportunity to improve our manuscript. To properly answer this question, we performed new experiments, namely isothermal entropy changes at different electric fields in DSC. This enabled us proving experimentally that $-T_s \Delta S_{\text{isothermal}}$ is very similar to $C_p \Delta T_{\text{adiab}}$ with C_p corresponding to the background value in our conditions.

Modification

The modification implemented below in the text is the same as the one we provided the first reviewer with in his fifth comment.

In the text (last paragraph in ‘Electrocaloric exchangeable heat in PST’

In this study, we consider that $C_p \Delta T_{\text{adiab}}$ stands for a good approximation of the exchangeable heat Q (Fig. 3a, right Y-axis). This assumption is often used in the literature [2, 29]. However, bulk PST transition is strongly 1st order and therefore exhibits a latent heat, which could induce some variations in our estimation. Consequently, we experimentally compared $C_p \Delta T_{\text{adiab}}$ and $-T_s \Delta S_{\text{isothermal}}$. Here C_p is considered as constant and obtained from DSC (background value in Fig 1b), ΔT_{adiab} is measured with the IR camera in adiabatic conditions and $\Delta S_{\text{isothermal}}$ comes from DSC measurements in which the electric field has been applied very slowly (200 s) to ensure isothermal conditions (Supp. Fig. 15a). At the lowest values of electric fields (11 and 14 kV cm⁻¹), which are the most sensitive ones because the transition is not fully driven, both values of heat are very similar (less than 5% apart) (Supp. Fig. 15b). This justifies the assumption of using $C_p \Delta T_{\text{adiab}}$ as a good estimation of Q in PST. Note also that considering the background value of C_p constitutes a lower bound of Q , meaning that η_{mat} cannot be overestimated with our method.

[29] S. G. Lu, B. Rožič, Q. M. Zhang, Z. Kutnjak, R. Pirc, Minren Lin, Xinyu Li, Lee Gorny, Comparison of directly and indirectly measured electrocaloric effect in relaxor ferroelectric polymers, Appl. Phys. Lett. 97, 202901 (2010)

Here is the text we suppressed from the initial version that has been replaced by the previous paragraph.

If one considers a constant C_p independent of field and temperature (300 J K⁻¹ kg⁻¹), a $\Delta T \approx 3.3$ K can be deduced from $\Delta T \approx T \Delta S / C_p$, equation systematically used in the literature, [2-3]. Here ΔS comes from our DSC isofield values at 18 kV cm⁻¹ and shown in Fig. 2c. This value is slightly higher than the one extracted from our IR measurements reaching 3.1 K (Fig. 3a). Consequently, in the following, we consider that ΔT_{adiab} enables obtaining a rather accurate estimation of the isothermal heat $Q = T \Delta S \approx \rho V C_p \Delta T_{\text{adiab}}$ (2), where ρ is the density and V is the volume (cf Fig. 3a). This allows the

simultaneous measurement of heat and electrical work, and thus the deduction of fairly accurate values of η_{mat} versus applied field.

In Supplementary Information (1)

In Supp. Section 3, we added the following text and modified Supp. Fig. 3d.

The measurements of C_p under electric field are quite challenging (Supp. Fig 3d.) because it is difficult to obtain a flat baseline. Indeed, the latter is affected by the connecting wires attached with silver paste to the sample. The results displayed in Supp. Fig. 3d are representative of all the measurements performed under electric field. With the accuracy of our set-up, we could not deduce any variation of C_p baseline versus electric field. Consequently, C_p equals $300 \text{ J kg}^{-1} \text{ K}^{-1} \pm 20 \text{ J kg}^{-1} \text{ K}^{-1}$, which is in line with literature [S4, S8]. The clear influence of electric field is C_p peak shifting towards higher temperature.

[S4] Crossley, S., Nair, B., Whatmore, R. W., Moya, X. & Mathur, N. D. Electrocaloric cooling cycles in lead scandium tantalate with true regeneration via field variation. *Phys. Rev. X* **9**, 041002 (2019).

[S8] Nair, B., Ph.D. thesis, University of Cambridge, 2020, available at <https://www.repository.cam.ac.uk/handle/1810/312805>

Supplementary Fig.3| **Temperature, latent heat, entropy change and specific heat deduced from isofield Differential Scanning Calorimetry (DSC) measurements on PST sample 1** a) Transition temperature of PST versus electric field while heating $T_{0,h}$ and cooling $T_{0,c}$, b) latent heat Q_0 versus electric field, c) entropy change ΔS_0 versus electric field, d) Specific heat C_p measurements under electric field.

In Supplementary Information (2)

The Supplementary Figure 15 below has been added in Supplementary Information, supporting the paragraph added in the main text.

Electrocaloric Exchangeable heat

Supplementary Fig. 15 | Comparison between $C_p \Delta T_{\text{adiab}}$ and $-T_s \Delta S_{\text{isothermal}}$. a) Isothermal application of an electric field of 11 kV cm^{-1} using DSC. It is applied very slowly (200 s) to maintain the temperature almost constant and measure a DSC signal. The integral under the DSC signal corresponds to the isothermal heat exchange $-T_s \Delta S_{\text{isothermal}}$. b) comparison of $C_p \Delta T_{\text{adiab}}$ to $-T_s \Delta S_{\text{isothermal}}$ at two electric field values (11 and 14 kV cm^{-1}). ΔT_{adiab} is measured with the IR camera in adiabatic conditions and C_p is considered as constant ($300 \text{ J kg}^{-1} \text{ K}^{-1}$), which corresponds to the background value of C_p measurements (figure 1b). The very good match between $C_p |\Delta T_{\text{adiab}}|$ to $-T_s \Delta S_{\text{isothermal}}$ proves that it is legitimate to consider $C_p \Delta T_{\text{adiab}}$ as an excellent estimation of the heat exchanged in η_{mat} .

c. PST exhibits a first-order phase transition which is accompanied by hysteresis losses. The authors need to comment on how the hysteresis losses in PST affect the materials energy efficiency (η_{mat}). This is critical when one compares the material's efficiency (η_{mat}) with Gd which exhibits a second-order phase transition with no hysteresis losses.

Answer: We thank the reviewer for this very important comment. We thought that the best way to answer this question is to consider a typical example of usage of this material in a device or prototype in which we would measure directly η_{mat} in a concrete example. We consequently ran specific experiments mimicking the working principle of a regenerator embedding bulk PST. We also ran finite element modelling on a proven model showing that this highly efficient material could be used in a regenerator. These extra experiments and simulations gave us the opportunity to define better the limitations of use of this material in an important concrete example.

Note that our answer here is the same as the one we provided the first reviewer with regarding his comment 2.b.

Modifications:

- In the article: The paragraph below has been added in the section *Discussion* (second paragraph).

Hence, the giant material efficiency observed in bulk PST stems from its first-order nature. A stringent associated constraint is that it occurs at well-defined temperatures. In the case of a realistic EC heat pump, it is legitimate to wonder how one could run a useful cycle and what the impact on η_{mat} would be. A first possibility is to use it in a 'one-shot' mode that is typically useful in electronic apparatus with a temperature threshold, as described in [25]. In this case, the maximum efficiency would be reached at the optimal temperature, meaning at 301 K according to Fig. 3a at 11 kV cm^{-1} , which is functional but quite limiting. A more suitable use is to integrate this material into a regenerator. The interest of such a device is to build a gradient of temperature larger than ΔT_{adiab} by running a specific thermodynamic cycle [4]. To do so, the positive ΔT_{adiab} must be triggered at a temperature higher than when the negative ΔT_{adiab} is triggered in this cycle. Otherwise, the built-in gradient would be destroyed. Moreover, both positive and negative ΔT_{adiab} s should exhibit the same magnitude to reach a steady state. We experimentally mimicked the working principle of such a device on our samples with a temperature-controlled hot plate stage, as depicted in Supp. Fig. 14. The idea is to build symmetrical cycles in the $\Delta T = f(T)$ map (Fig 3a and Supp. Fig. 14c). For instance, at 11 kV cm^{-1} , it is possible to obtain identical positive and negative $|\Delta T_{\text{adiab}}| = 1.7 \text{ K}$ if we trigger them respectively at 300.5 K and 300 K (Regenerator A, Supp. Fig. 14a). At such values, we measured η_{mat} of at least 92. The price to pay is that Q decreases (4.63 J cm^{-3}). Other functional examples are given in Supp. Info. 14. Moreover, we checked that bulk PST can be run in a regenerator by simulating it with a proven finite element model, detailed in [4] (Supp. Fig. 14.1). Hence, although the constraints imposed by PST first-order nature are rather strong, it is possible to find experimental conditions enabling taking advantage of PST bulk giant η_{mat} .

[4] Torello, A. *et al.* Giant temperature span in electrocaloric regenerator. *Science* **370**, 125-129 (2020).

[25] Defay, E., Mathur, N., Kar-Narayan, S., Soussi, J., Method for limiting the variation in the temperature of an electrical component, **US9326423B2**, 2012.

- In Supplementary: Simulations and experiments discussed in the previous paragraph have been added in Supplementary 14.

14 - Potential Application of bulk PST

We show below how our PST could be used in a fluid-based regenerator without being affected by the asymmetry of ΔT_{adiab} , irreversibility or hysteresis losses. From ΔT_{adiab} measurements (Supp. Fig.14c) one can define a window of temperature where the regenerator could operate. At the maximum materials efficiency obtained at 11 kV cm^{-1} , we could simulate experimentally reproducible and reversible regenerators (Supp. Fig.14a and 14b). The regenerator would operate in a small temperature span at a lower heat exchanged (Supp. Fig. 14d). The heat can be increased by increasing the applied electric field but the materials efficiency will decrease as the heat saturates while the electrical work increases (Supp. Fig 14.2).

Furthermore, using the ΔT_{adiab} measurements in Supp. Fig.14c, we did some simulations of a fluid-based regenerator based on a similar model published in [S13] operating at different starting temperatures around room temperature and without any kind of heat losses to the environment. We show that a temperature gradient can be reached in all cases after several cycles (Supp. Fig 14.1). This indicates that, despite an asymmetry in ΔT_{adiab} or hysteresis losses, bulk PST could be used to build different kinds of regenerator prototypes.

[S13] Torello, A. *et al.* Giant temperature span in electrocaloric regenerator. *Science* **370**, 125-129 (2020).

Supplementary Fig. 14 | **Potential use of PST bulk in a regenerator.** Here we show two potential working points of PST bulk if it were integrated in a regenerator. The cycle of the latter has been mimicked with a temperature-controlled hot plate stage (Linkam). In a (resp. b), PST is first set at 300.5 K (resp. 301 K). The EC positive $\Delta T_{\text{adiab}} = 1.7$ K (resp. 1.4 K) is then triggered by charging PST. Heat is exchanged and PST goes back to 300.5 K (resp. 301 K). If PST was in a regenerator, a fluid (for instance) would then be displaced and PST temperature would decrease. Here, we suppose that this temperature is 300 K (resp. 299.5 K). This value is chosen in order to obtain a symmetrical position in $\Delta T_{\text{adiab}} = f(T_s)$ displayed in c. This EC negative $\Delta T_{\text{adiab}} = -1.7$ K (resp. -1.4 K) is then triggered by discharging PST and PST exchanges heat until it reaches 300 K (299.5 K) again. And the cycle carries on as it would in a proper regenerator. **a)** Regenerator A operating between 300-300.5 K, on a temperature window of 0.5 K. **b)** Regenerator B operating on a temperature window of 1.5 K. **c)** Adiabatic temperature change of PST bulk at 11 kV cm^{-1} . The orange and black lines represent respectively the regenerators A and B. **d)** the table shows for each regenerator, its temperature window, the heat exchanged and the corresponding measured materials efficiency.

Supplementary Fig. 14.1 | **Regenerator modelling.** The time evolution of the hot side (red) and cold side (blue) of an active regenerator based on the EC effect from Supp. Fig.14c at starting temperatures **(a)** 297 K, **(b)** 299 K, **(c)** 301 K and **(d)** 303 K. **(e)** shows the dimensions of the regenerator simulated. The simulation consists of a finite element method (FEM) 2D representation of an active regenerator made with a single PST plate of 0.2 mm x 4 cm. No losses to the surroundings were considered. All the parameters of this model are detailed in [4] [S13].

Supplementary Fig. 14.2| **Potential regenerator for PST bulk at higher heat Q.** The heat exchanged can be increased by increasing the electric field but this will decrease the materials efficiency as shown in **d**). **a**) regenerator C at the electric field of 14 kV cm⁻¹ **b**) regenerator D at 18 kV cm⁻¹ **c**) regenerator E at field of 22 kV cm⁻¹ **d**) for each regenerator, the heat exchanged, materials efficiency and adiabatic temperature change.

d. During the IR camera measurements of reversible adiabatic temperature change, the effect of Joule heating due to leakage currents in the sample needs to be carefully taken into consideration. The authors mention that "This was done without Joule heating as the initial temperature of the sample was recovered after application of maximum field 22 kV cm⁻¹." However with sufficient time the initial temperature of the sample is likely to be recovered so the statement needs to be revised. I recommend providing some of the actual IR images of the PST sample during field cycling in the Supplementary Information. Further leakage currents should be measured to make sure that the direct measurements of the adiabatic temperature changes by IR imaging are not affected by extrinsic factors.

Answer: We thank the reviewer for this important comment. As recommended, we performed extra experiments to collect actual IR images of the PST sample during field cycling in Supplementary Information. No Joule effect could be spotted on collected temperature variations. Moreover, we measured leakage current in the sample under electric field in order to estimate how much this leakage could influence our measurement of electric work. It turns out that it is at maximum 1%, which is negligible.

Modifications:

In the paper

We added a specific paragraph “Joule heating assessment” at the end of the “Methods” section, in which we gathered the previous comments about Joule effect that were in the Methods/IR camera paragraph and the new ones stemming from our new measurements.

Joule heating assessment

We measured a stable leakage current $I_{\text{leak}} = 0.2 \mu\text{A}$ when maximum voltage (1100V) was applied during the thermalization phase of bulk PST sample 1. This induces a Joule heating power of 0.2 mW, which represents less than 30 μJ of Joule heating during the 0.12 s of the adiabatic step. In Fig. 3c, W_e is always beyond 2 mJ when 1100 V have been applied (corresponding to 22 kV cm^{-1} in our sample). Therefore, the influence of Joule heating in W_e is less than 1.5 %, which can be considered as negligible. Besides, we could not observe any influence of Joule heating in the IR camera characterization. In Supp. Fig. 9.1, the stable temperature after thermalization following the application or the removal of the electric field is the same in both cases. We also display in Supp. Fig. 9.5 the IR images of PST sample 1 during a standard IR sequence, namely stable initial temperature, adiabatic heating step (field on), thermalization (field on), cooling step (field off), thermalization (field off), in which the colours displayed by thermalized steps – representative of sample’s temperature - are the same as the initial step. This means that Joule heating also plays a negligible role in the determination of ΔT_{adiab} – and therefore of heat - in our experiments.

In supplementary

IR images of the bulk ceramic PST during field cycling have been added to Supplementary Information (Supp. Fig. 9.5).

Supplementary Fig. 9.5 | IR images scans of bulk PST sample 1 during a standard EC characterization sequence. In step 1 the material is at starting temperature T_s of 299 K, in step 2, 1100 V are applied and the materials temperature increases. Subsequently, in the two following images the material thermalizes and goes back to T_s (no Joule heating). In step 4, the materials temperature decreases as voltage is removed. Finally, in Step 5 the material thermalizes back to T_s .

e. Table 1 and table 2 are not separately provided. This needs to be corrected.

Answer: Both tables are now separately provided.

f. The authors claim that the high efficiency observed in the PST sample is due to its high degree of ordering. However, the authors use only powder XRD for calculating the degree of ordering in PST

which could have some errors. It will be beneficial to the readers if the authors can cite some references in the text in this regard.

Answer and modification: This is indeed customary in the literature to characterize PST degree of ordering by powder XRD. As recommended, we added two references in the Method section detailing the calculation of the degree of ordering in PST. Both references are also mentioned in Supplementary Information 1.

X-Ray Diffraction measurements: XRD diffraction was carried out with PANalytical X'Pert Pro on the powdered PST from 15 to 60 ° angles at every 0.02 °. The B-site cation order Ω was computed from the integrated intensities of the X-ray diffraction maxima 111 and 200 (Supplementary Fig.1) following the standard technique detailed in [11] and [28].

[11] Shebanov, L., Birks, E. H. & Borman, K. X-ray studies of electrocaloric lead scandium tantalate ordered solid solutions. *Ferroelectrics* **90**, 165-172 (1989).

[28] Wang, H. and Schuize, W.A. Order-Disorder Phenomenon in Lead Scandium Tantalate. *J. Am. Ceram. Soc.* **73**, 1228 (1990).

REVIEWER COMMENTS

Reviewer #1 (Remarks to the Author):

My questions and comments are addressed by the authors. Thanks for the clarification. The revised manuscript is acceptable for publication.

Reviewer #2 (Remarks to the Author):

Authors resolved the issues from reviewers. However, from the basic materials' view point, there is one critical issue to be clarified clearly.

In a SI Fig. 12 for PE loop, PST transforms from ferro to anti-ferro (at 299K) to para. This phase transition is quite interesting. What is the mechanism for this unusual phase transition? Is that phase transition is related with high electro-caloric effect?

Ferroelectric Hysteresis is related with loss so that many EC materials were tuned to have a para-like properties. For example, (BaSr)TiO₃ is one of typical example. To my knowledge this large loss (hysteresis) may lead to the lower efficiency. However, in this experiment, it seems that Ordering enhanced the ferroelectricity and larger ΔK ? Why?

- J. Appl. Phys. 102, 044110 (2007); <https://doi.org/10.1063/1.2770834>

It seems that The peak position in SI Fig. 3d with 0 E-field is different from the peak in Fig. 1 b. why? In addition, SI Fig. 3 d caption should mention that these curves are from cooling or heating.

- J. Appl. Phys. 102, 044110 (2007); <https://doi.org/10.1063/1.2770834>

It seems that The peak position in SI Fig. 3d with 0 E-field is different from the peak in Fig. 1 b. why? In addition, SI Fig. 3 d caption should mention that these curves are from cooling or heating.

Figure A - PE loops on PST bulk ceramic from literature with a double loop hysteresis. a) figure from Shebanov et al., *Ferroelectrics* **184**, 239–242 (1996). b) is taken from Crossley, S. et al., *Phys. Rev. X* **9**, 041002 (2019)

As anticipated by the reviewer, this phase transition is indeed responsible for the high electrocaloric effect. Hence, Shebanov wrote in [Shebanov et al., *Ferroelectrics* **184**, 239–242 (1996)] that PST experiences a “field-induced first order phase transition manifested by the appearance of double dielectric hysteresis loops” that “provides a dominant contribution to the EC effect”.

Regarding the presence of double loops, we can also mention our work on lead zirconate (PZO), a true antiferroelectric material. In PZO, antiferroelectricity infers a very peculiar electrocaloric effect. In [Vales-Castro, P. et al, *Phys.Rev. B* **103**, 054112 (2021)] we observed double loops in PZO ceramics, somehow similar to the ones in PST. We showed though that by increasing temperature, PZO goes sequentially from antiferroelectric to ferroelectric and then to paraelectric phase. These successive phase transitions can be driven by an external electric field that gives rise to an outstanding negative electrocaloric effect when PZO goes from antiferro to ferro, a transition that has not been observed in PST.

Modifications

- In the main text, we mentioned that the phase transition can be driven by an electric field and we added Shebanov’s paper reference at the end of the sentence. In the upper part of page 3, it now reads: “It exhibits a first-order ferroelectric to paraelectric phase transition near room temperature which makes it attractive for air conditioning. **This transition can also be driven with an electric field leading to the EC effect, which is predominant at the material’s transition temperature [13].**”
- In Supplementary Information (SI) section 12, supplementary PE loops of PST have been added at 303 K and 305 K in addition to the ones already present in Fig. SI 12. The following explanation has also been added along with new references in section 12 of SI.

As shown in SI Fig. 12, PST is ferroelectric at 288 K and paraelectric at 313 K. Between 299 K and 305 K (SI Fig. 12) we observe double loops, which suggests antiferroelectricity. However, as already observed and explained by Shebanov first [S5] and then by Crossley [S12], these double loops are evidences of a paraelectric to ferroelectric electric field-driven phase transition. Note that it was observed in 1953 in BaTiO₃, another electrocaloric material with a first order phase transition [S14]. This phase transition is responsible for the high electrocaloric effect observed. Hence, Shebanov wrote that PST experiences a “field-induced first order phase transition manifested by the appearance of double dielectric hysteresis loops” that “provides a dominant contribution to the EC effect” [S12].

Regarding the presence of double loops, the EC properties of lead zirconate (PZO), a true antiferroelectric material, can be mentioned and compared with PST. In PZO, antiferroelectricity infers a very peculiar electrocaloric effect. In [S15], we observed double loops in PZO ceramics, somehow similar to the ones in PST. We showed though that by increasing temperature, PZO goes

sequentially from antiferroelectric to ferroelectric and then to paraelectric. These successive phase transitions can be driven by an external electric field that gives rise to an outstanding negative electrocaloric effect when PZO goes from antiferroelectric to ferroelectric, a transition that has not been observed in PST.

[S5] Shebanov, L., Sternberg, A., Lawless, W. N. & Borman, K. Isomorphous ion substitutions and order-disorder phenomena in highly electrocaloric lead-scandium tantalate solid solutions. *Ferroelectrics* **184**, 239–242 (1996).

[S12] Crossley, S., Mathur, N.D. & Moya, X. New developments in caloric materials for cooling applications. *AIP Advances* **5**, 061753 (2015).

[S14] Merz, W.J., Double hysteresis loop of BaTiO₃ at the Curie point. *Phys. Rev.* **91**, 513 (1953)].

[S15] Vales-Castro, P. et al, *Phys.Rev. B* 103, 054112 (2021)

Supplementary Fig. 12 | **Polarisation-electric field loops of 0.5 mm-thick bulk PST sample 1 at different temperatures (from 288 K to 313 K).** The measurements were done using a standard Sawyer-Tower circuit at 20 Hz up to 18 kV cm⁻¹. The blue area is $W = \int E dD$, which represents the electrical work per volume unit to charge PST at 299 K up to 18 kV cm⁻¹.

2. *Ferroelectric Hysteresis is related with loss so that many EC materials were tuned to have a para-like properties. For example, (BaSr)TiO₃ is one of typical example. To my knowledge this large loss (hysteresis) may lead to the lower efficiency.*

Answer We thank the reviewer for his/her comment giving us the opportunity to give more details about this important question.

Ferroelectric hysteresis infers losses. They have *de facto* been considered in our calculation of materials efficiency because we measured the current needed to charge PST, which we called direct measurement, contrary to extracting electrical work from *PE* loops). Our method is equivalent to considering not only the hysteresis losses but the entire work needed to fully charge the capacitor, as represented by the blue shaded area in Figure B below representing the *PE* loops of PST at 299 K. More specifically, our experimental extraction is equivalent to using the lower branches of the *P*(*E*) loops to calculate the electrical work $W = \int E dD$ (cf Figure B). Note that we used the term “*PE*

loops” though these loops are “ DE loops”, D being the electrical displacement field. However, this is standard practice to consider P as being equivalent to D in the case of high dielectric constant materials such as PST.

Figure B – The blue shaded area corresponds to the electrical work considered to calculate materials efficiency, thereby all the hysteresis losses are included in our calculation.

To clearly confirm that our calculation is equivalent to extracting electrical work from $\int EdD$ in PE loops, we compared the electrical work that we extracted with 1) our direct method in which PST is charged at constant current up to a fixed voltage and 2) from PE loops up to the same voltage ($\int EdD$) as explained in Figure B. The results are depicted in Figure C and show that both methods give very similar results as expected. This is no surprise as both methods essentially involve the same extraction, P being an integration of the current in PST while charging.

Consequently, hysteresis losses are fully taken into account in our materials efficiency.

Figure C – Electrical work needed to charge PST up to 18 kV/cm from the direct method (charge at constant current) and from integrating EdD from a DE loop.

Modifications

- In supplementary information, section 12, we added the following discussion and figure.

Ferroelectric hysteresis infers losses. They have *de facto* been considered in our calculation of materials efficiency because we measured the current needed to charge PST. Our method is equivalent to considering not only the hysteresis losses but the entire work needed to fully charge

the capacitor. More specifically, our experimental extraction is equivalent to using the lower branches of the $P(E)$ loops to calculate the electrical work $W = \int E dD$, as depicted by the blue shaded area of the PE loop at 299 K in supplementary Figure 12. Note that we used the term “ PE loops” though these loops are “ DE loops”, D being the electrical displacement field. However, this is standard practice to consider P as equivalent to D in the case of high dielectric constant materials such as PST. Supplementary Figure 12 bis represents the electric work W_e extracted respectively from the direct method (same as Fig. 3c) and the PE loop method for an electric field of 18 kV cm^{-1} . It shows that both methods give very similar results. This is no surprise as both involve current measurements, P being an integration of the current in PST while charging. Consequently, hysteresis losses are fully taken into account in our materials efficiency.

Supplementary Figure 12 bis – Electrical work needed to charge PST up to 18 kV/cm from the direct method (charge at constant current) in black and from integrating EdD from a PE loop in red.

3. However, in this experiment, it seems that Ordering enhanced the ferroelectricity and larger delta K ? Why? - *J. Appl. Phys.* 102, 044110 (2007); <https://doi.org/10.1063/1.2770834>

Answer As already observed in 1996 by Shebanov [Shebanov, et al., *Ferroelectrics* 184, 239 (1996)] on bulk PST and confirmed in our paper (Supplementary Fig. 6 below and Table 1), increasing ordering strongly enhances the EC effect (larger ΔT_{adiab}). The paper suggested by the reviewer (*J. Appl. Phys.* 102, 044110 (2007)) is about the impact of ordering on the dielectric constant in thin films but there is no mention of the influence on the EC effect. In the referenced paper, PST dielectric constant increases with ordering whereas this is the opposite in ceramics, as reported by Shebanov in Shebanov et al., *Ferroelectrics* 184, 239–242 (1996). Besides, note that we reported in supplementary Figure 6 (reported below for convenience) the impact of ordering in bulk PST on the EC effect, collecting information from several publications and our work. The trend is very clear, namely the larger the ordering, the higher the EC effect.

Supplementary Figure 6| **Adiabatic temperature change of bulk ceramics PST**. The bulk ceramics PST collected from literature [S3] [S4] [S5] are compared to our PST samples (Sample 1, $\Omega=0.98$ and Sample 2, $\Omega=0.89$). ΔT_{adiab} of sample 1 data is shown in Supp. Info 10 and ΔT_{adiab} of sample 2 are presented in Supp. Info 5. The number displayed in bold closed to each symbol is the field applied in kV cm^{-1} . In brackets the references.

4. It seems that the peak position in SI Fig. 3d with 0 E-field is different from the peak in Fig. 1 b. why?

Answer We thank the reviewer for his/her comment that helps to improve our manuscript.

Figure 1.b and SI Fig. 3d come from two different experiments. In Fig 1.b the DSC measurements were carried out on a naked sample with no electrodes and wires. In SI Fig. 3d, the DSC measurements were carried out on a wired sample with a layer of electrodes on both bottom and top surfaces. The wires and electrodes induce an inferior thermal contact leading to a slight temperature shift of the peak at zero field compared to Figure 1b. However, the latent heat (integral under the peak) is the same in fig 1b (1031 J kg^{-1}) and SI Fig 3d (1037 J kg^{-1}).

Modifications

- In Supplementary: the text below has been added in section 3 to explain the difference in the peak positions.

Here the measurements were carried out on a wired sample attached with silver paste (electrodes). The wires and electrodes induce an inferior thermal contact leading to a slight difference in the peak at zero field compared to Figure 1b. However, the latent heat (integral under the peak) is the same in Fig 1b (1031 J kg^{-1}) and SI Fig 3d (1037 J kg^{-1}).

5. In addition, SI Fig. 3 d caption should mention that these curves are from cooling or heating.

Answer We thank the reviewer for his/her comment. We now mention in SI Fig.3d that these curves have been collected while heating.

Modifications

SI Fig. 3d d) Specific heat C_p measurements under electric field on heating.

REVIEWERS' COMMENTS

Reviewer #2 (Remarks to the Author):

Authors resolved the issues from reviewers.

Answers to the reviewers

Italic – comments from reviewers

Standard text – answers to the reviewers

Highlighted yellow – modifications implemented in the text

Reviewer #2 (Remarks to the Author):

Authors resolved the issues from reviewers.

Answer: We are thankful to the reviewer for having helped us improve our manuscript.